# F-Adapter: Frequency-Adaptive Parameter-Efficient Fine-Tuning in Scientific Machine Learning

**Hangwei Zhang[1,2,3]    Chun Kang[2,3]    Yan Wang[2†]    Difan Zou[1†]**

[1] School of Computing and Data Science, The University of Hong Kong
[2] Institute for AI Industry Research, Tsinghua University
[3] Beihang University

{hangweizhang,kangchun}@buaa.edu.cn, wangyan@air.tsinghua.edu.cn, dzou@cs.hku.hk

## Abstract

Parameter-efficient fine-tuning (PEFT) of powerful pre-trained models for complex downstream tasks has proven effective in vision and language processing, yet this paradigm remains unexplored in scientific machine learning, where the objective is to model complex physical systems. We conduct the first systematic study of PEFT for pre-trained Large Operator Models (LOMs) obtained by scaling variants of Fourier Neural Operator. First, we observe that the widely used Low-Rank Adaptation (LoRA) yields markedly poorer performance on LOMs than Adapter tuning. Then, we further theoretically establish that stacked LoRA incurs a depth-amplified lower bound on approximation error within Fourier layers, whereas adapters retain universal approximation capacity and, by concentrating parameters on energy-dominant low-frequency modes, attain exponentially decaying error with bottleneck width in the Fourier domain. Motivated by the robust empirical gains of adapters and by our theoretical characterization of PDE solutions as spectrally sparse, we introduce Frequency-Adaptive Adapter (F-Adapter). F-Adapter allocates adapter capacity based on spectral complexity, assigning higher-dimension modules to low-frequency components and lower-dimension modules to high-frequency components. Our F-Adapters establish state-of-the-art (SOTA) results on multiple challenging 3D Navier–Stokes benchmarks, markedly enhancing both generalization and spectral fidelity over LoRA and other PEFT techniques commonly used in LLMs. To the best of our knowledge, this work is the first to explore PEFT for scientific machine-learning and establishes F-Adapter as an effective paradigm for this domain. The code is publicly available at here.

## 1  Introduction

Learning solution operators for partial differential equations (PDEs) is a fundamental challenge in scientific machine learning (SciML). Among the most promising approaches are operator-learning architectures, particularly the Fourier Neural Operator (FNO) and its variants [23, 52, 49, 28, 3, 12, 24]. These models leverage mesh-independent spectral convolutions to efficiently capture fine-scale dynamics in the frequency domain [11, 38, 55], enabling orders-of-magnitude faster inference compared to traditional numerical solvers [37, 2]. Recently, the field has seen the rise of Large Operator Models (LOMs) [61], which scale these architectures and employ large-scale pre-training on diverse datasets, unlocking remarkable generalization capabilities for complex downstream tasks.

When adapting large pretrained models to downstream tasks, parameter-efficient fine-tuning (PEFT) has emerged as a powerful strategy, offering minimal computational and storage overhead [27, 56, 29, 21, 13]. Unlike full-model fine-tuning, PEFT techniques fine-tune only a small subset of trainable parameters. This approach preserves the benefits of pretraining while enabling rapid deployment

39th Conference on Neural Information Processing Systems (NeurIPS 2025).

across tasks and domains. This paradigm has proven highly effective in natural language processing (NLP) [59, 26, 36] and computer vision (CV) [32, 5, 19], where large foundation models dominate and efficient task adaptation is critical for scalability.

However, despite its effectiveness in NLP and CV, its potential within SciML remains unexplored. Although a few recent studies employ LoRA-style Physics-Informed Neural Networks (PINNs) [40] to build surrogates for parameterized PDEs [7, 51], they train modest networks from scratch on small and single-type equation datasets. A systematic PEFT study for pre-trained LOMs therefore remains to be established. Physical systems governed by PDEs pose qualitatively different challenges: their solution manifolds exhibit broadband, cascade-coupled spectra and reside in high dimensional continuous domains [35, 34, 22]. These distinctions prompt our central question: Can PEFT be adapted to LOMs in SciML so that it explicitly respects the frequency-adaptive structure and physics-based priors inherent to PDE solution spaces?

In this work, we present the first systematic study of PEFT for pretrained LOMs. Through a combination of empirical analysis and theoretical investigation, we identify a fundamental limitation in the widely used LoRA approach [18]: its rank-constrained linear updates create a depth-amplified spectral error floor when applied to Fourier-based operator architectures. On the other hand, we show that replacing these linear updates with lightweight *non-linear* adapters, implemented as residual two-layer MLP bottlenecks, can lead to surprisingly effective fine-tuning for LOMs. We further demonstrate that this approach can maintain universal approximation capabilities while strategically concentrating model capacity on the energy-dominant spectral subspace, thus enabling parameter-efficient adaptation without sacrificing spectral fidelity [17].

Building on these insights, we propose Frequency-Adaptive Adapters (F-Adapters), a novel PEFT architecture for LOMs that allocates adapter capacity according to spectral complexity. Concretely, the Fourier Layer in LOMs bins its Fourier coefficients into different spaced radial shells, creating disjoint frequency bands for capacity-aware F-Adapter assignment. Specifically, F-Adapters assign larger bottleneck dimensions to low-frequency bands which typically contain most of the signal energy and govern long-range physical interactions, and smaller dimensions to high-frequency bands that often sparse and susceptible to numerical noise. We summarize our main contributions as follows:

- We empirically and theoretically establish that residual two-layer MLP *Adapters* significantly outperform LoRA for fine-tuning in scientific machine learning. Next, we rigorously analyze the energy distribution of PDE solutions in the Fourier domain. All the resulting theory guides the design of our architectural innovations.

- We devise a **Frequency-Adaptive Adapter** (F-Adapter) that allocates parameters in proportion to the spectral energy profile of PDE operator solutions, which in turn couples model capacity to task-relevant frequencies.

- We achieve the SOTA performance on multiple challenging 3D Navier–Stokes forecasting benchmarks, which surpasses LoRA and prior PEFT baselines in L2RE accuracy with only less than 2% of backbone parameters tuned. Comprehensive ablation studies and direct comparisons with other Fourier domain adapter designs confirm the superior effectiveness of F-Adapters.

## 2 Related Works

**Parameter-Efficient Fine-Tuning.** PEFT adapts frozen backbones through minimal trainable components. Prompt Tuning learns a compact "soft" prompt that is prepended to the input while keeping all model weights fixed [20]. Adapter tuning inserts narrow bottleneck MLPs between Transformer sub-layers so that only these adapters are updated [17]. FiLM Adapter extends adapter tuning by treating channel-wise FiLM layers as adapters and updating only their $(\gamma, \beta)$ parameters [43]. LoRA injects a pair of trainable low-rank matrices whose product is added to each frozen weight tensor [18]. AdaLoRA allocates the low-rank budget across layers dynamically according to data-driven importance scores [58]. HydraLoRA shares one down-projection across multiple LoRA heads, enlarging expressiveness without extra memory [48]. RandLoRA couples fixed random bases with trainable diagonal scalings to approximate full-rank updates at constant parameter cost [1]. SVFT updates each tensor via a sparse mixture of its own singular-vector outer products, training only the corresponding coefficients [25]. Concurrently, Loeschcke et al. [31] uses Tucker-factorized low-rank updates, preserving cross-mode structure and outperforming unfolding-

based LoRA. Together, these methods illustrate how structural priors can drastically reduce trainable parameters while retaining fine-tuning flexibility.

**Pretrained Large Operator Models for PDE Solving.** A rapidly growing body of work now treats LOMs as foundation models. These models are first pretrained on heterogeneous collections of partial differential equations and are later adapted to new physical regimes. The pioneering study of Subramanian et al. [45] shows that a FNO trained on eight disparate PDEs scales predictably and slashes downstream data needs by orders of magnitude. Building on this, MPP adds an autoregressive transformer over ten systems for strong zero-shot transfer [33], Poseidon cuts cost via multiscale conditioning [16], UPS employs cross-modal adaptation for data-efficient generalisation [42], and CoDA-NO introduces codomain-aware attention for few-shot multiphysics tasks [39]. PreLowD [15] and OmniArch [6] demonstrate the powerful generalisation capabilities of large-scale operator models achieved by moving spatial field values into the frequency domain. DPOT further scales to 1B parameters with Fourier-denoising pretraining, achieving SOTA on $10+$ datasets [14]. Collectively, these LOMs demonstrate that heterogeneous PDE pretraining provides a powerful tool for scientific machine learning.

# 3 Behavior of Fine-Tuning Methods for Large Operator Models

In this section, we examine the performance of fine-tuning techniques applied to pretrained Large Operator Models (LOMs). First, we conduct an empirical comparison of some typical fine-tuning methods, followed by a theoretical interpretation of the results. Next, we delve into a deeper theoretical analysis to illustrate how the information within the solutions is distributed across Fourier spaces. This analysis offers valuable insights that can inform the development of more effective PEFT methods for LOMs.

## 3.1 Empirical Comparions between Different Fine-Tuning Methods for LOMs

**Task and experiment setup.** To evaluate fine-tuning methods for LOMs, we focus on the three-dimensional forecasting problem, a challenging scientific machine learning task characterized by a highly nonlinear high-dimensional solution manifold and unstable truncation errors. Specifically, we use DPOT-H as the pretrained model, a 1B parameter backbone that is currently the largest publicly available LOM [14]. We fine-tune this model on two 3D Navier–Stokes datasets from PDEBench [46] with standard parameter-efficient methods, including LoRA and bottleneck adapters (implementation details appear in Appendix A). These datasets are configured with random initial conditions at $M = 1.0$ and $M = 0.1$.

We primarily integrate the PEFT modules into the Fourier-Attention layers, which form the computational core of the model, contain the majority of its parameters, and dominate the overall computational cost. During fine-tuning, we use the AdamW optimizer and train the model for 500 epochs with different efficiency levels (e.g., ranks for LoRA and bottleneck dimensions for Adapter). The performance is

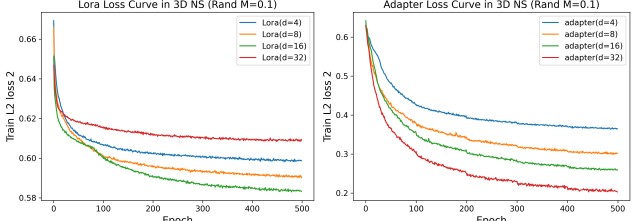

Figure 1: Convergence comparison of LoRA and bottleneck Adapter. Adapter not only starts with a lower loss but also reaches a lower steady-state value, indicating faster and more stable convergence.

evaluated on the test set using the *L2 relative error* (L2RE), a standard metric in operator learning [23]. All experiments are conducted on a single NVIDIA A800 80 GB GPU. Complete experimental details are provided in Appendix C.

**Experimental results.** Building on the experimental setup outlined earlier, we conducted a comprehensive empirical comparison of the performance of the LoRA and Adapter methods across varying ranks and bottleneck dimensions. The results are summarized in Figure 1 and Table 1, where several key insights can be revealed. First, despite the widespread adoption of the LoRA method in many large language model (LLM) tasks, it demonstrates significantly poorer performance in fine-tuning LOMs (low-rank models) and does not benefit from increasing the ranks. Second, the Adapter method proves highly effective in fine-tuning LOMs (original models). The performance continues to

improve as the bottleneck width increases, although larger width introduces some overfitting. These findings highlight the distinct suitability of each method depending on the specific model architecture and fine-tuning objectives. This suggests that **the Adapter method may be a more appropriate choice for fine-tuning LOMs in modeling physical systems.**

| Scheme | % Params | Mem (GB) | L2RE ($M$=1.0) | L2RE ($M$=0.1) |
|---|---|---|---|---|
| LoRA ($r$=4) | 0.17% | 12.58 | 0.6413 | 0.6218 |
| LoRA ($r$=8) | 0.34% | 12.65 | 0.6345 | 0.6129 |
| LoRA ($r$=16) | 0.69% | 12.78 | 0.6427 | 0.6147 |
| LoRA ($r$=32) | 1.37% | 15.85 | 0.6395 | 0.6211 |
| Adapter ($d$=4) | 0.59% | 15.82 | 0.6169 | 0.5063 |
| Adapter ($d$=8) | 1.16% | 15.85 | 0.5496 | 0.4893 |
| Adapter ($d$=16) | 2.30% | 15.89 | 0.5227 | **0.4539** |
| Adapter ($d$=32) | 4.59% | 15.98 | **0.5134** | 0.4570 |

Table 1: Comparison on LoRAs with different rank and Adapters with different bottleneck dimension.

## 3.2 Theoretical and Empirical Explanations on the Benefit of Adapter Methods

In this part, we further provide explanations about why adapter can lead to substantially better performance than LoRA in fine-tuning LOMs.

We mainly consider the comparison between the **Block-wise LoRA** and **two–layer MLP Adapter** within the Fourier layers of LOMs. In particular, block-wise LoRA applies the multiple low-rank adaption for different blocks of the target parameters separately. For adapter model, we model the adapter in Fourier blocks as a two-layer MLP $g : \mathbb{C}^N \to \mathbb{C}^N$, $g(\hat{x}) = U\, \sigma(V\hat{x} + b) + c$, with weights $V \in \mathbb{C}^{m \times N}$, $U \in \mathbb{C}^{N \times m}$, biases $b \in \mathbb{C}^m$, $c \in \mathbb{C}^N$, and non-linearity $\sigma(\cdot)$.

Then, we first deliver the following proposition that characterizes the approximation error for the block-wise LoRA method with rank $r$.

**Proposition 3.1** ( Block-wise LoRA lower bound ). *Let $\Delta W_{\mathrm{g}} = \mathrm{blockdiag}\big(\Delta W^{(1)}, \ldots, \Delta W^{(K)}\big)$ be the block-wise model parameter updates and $BA = \mathrm{blockdiag}\big(B^{(1)}A^{(1)}, \ldots, B^{(K)}A^{(K)}\big)$ be the block-wise low-rank approximation, where $B^{(k)} \in \mathbb{C}^{d \times r}, A^{(k)} \in \mathbb{C}^{r \times d}$. Then, for any input $x$, the approximation error for block-wise LoRA satisfies*

$$\big\|(\Delta W_g - BA)x\big\| \;\geq\; \left(\sum_{k=1}^{K} \sum_{i=r+1}^{d} \sigma_{k,i}^2\, (v_{k,i}^\top x_k)^2\right)^{1/2}. \tag{1}$$

*In particular, the worst-case operator-norm error obeys*

$$\sup_{\|\hat{x}\|_2 = 1} \|(\Delta W_g - BA)\hat{x}\|_2 \;\geq\; \sigma_{Kr+1}(\Delta W_g) \tag{2}$$

**Interpretation.** Even if each block is well approximated in isolation, the *worst-case* LoRA error across the entire stack is still lower-bounded by the $(Kr+1)$-th singular value of the global matrix, revealing an intrinsic **additive bottleneck** as depth $K$ grows.

Adapters, on the other hand, introduce non-linear, width-controlled bottlenecks *after* the Fourier transform. Rather than adjusting the pre-trained model's parameters, it carries out fine-tuning in a separate representation space. Because the underlying two-layer MLP satisfies the universal-approximation theorem, it is, in principle, capable of representing any measurable function. In the Fourier layers of large operator models (LOMs), however, the practical rate at which this universal approximation is achieved is dictated by the spectral frequency content of the target update. We first introduce the following notation for adapters: Let $x \in \mathbb{R}^{d_1 \times d_2 \times d_3}$ and define the unitary discrete Fourier transform and its inverse $\mathcal{F} : \mathbb{R}^{d_1 \times d_2 \times d_3} \to \mathbb{C}^N$, $\mathcal{F}^{-1} : \mathbb{C}^N \to \mathbb{R}^{d_1 \times d_2 \times d_3}$, $N = d_1 d_2 d_3$. For a multi-index $k \in \mathbb{Z}^3$ we abbreviate $\langle k \rangle := \sqrt{1 + \|k\|^2}$. An Adapter's Fourier coefficients are defined by $g_k = \frac{1}{(2\pi)^d} \int_{\mathbb{T}^d} g(x)\, e^{-ik \cdot x}\, dx$, where $d = d_1 d_2 d_3$, $k \in \mathbb{Z}^d$ and $\mathbb{T}^d$ is the $d$-dimensional torus.

**Proposition 3.2** (Frequency-selective approximation of adapters). *Let $|g_k| \leq C\langle k \rangle^{-\alpha}$ with $\alpha > \frac{d}{2}$. For any $\varepsilon > 0$ there exist frequency truncation radius $K > 0$ and adapter bottleneck width $m \in \mathbb{N}$ such that the Fourier-domain adapter $\hat{g}$ obeys*

$$\|e\| := \|\mathcal{F}^{-1}(g) - \mathcal{F}^{-1}(\hat{g})\|_2 < \varepsilon, \qquad \|e\| = O\big(K^{\frac{d}{2} - \alpha}\big) + O\big(K^{\frac{d}{2}} e^{-cm}\big). \tag{3}$$

**Implications.** All proofs are given in Appendix B. Frequency-agnostic linear *Block-wise LoRA* is bottle-necked by the depth-dependent lower bound in Proposition 3.1, whereas *Adapters* in Fourier blocks concentrate parameters on the energy-dominating low-frequency subspace and exploit the exponential accuracy of Proposition 3.2. This frequency-selective compression accounts for the superior balance between predictive accuracy and parameter count observed in all the experiments on DPOT [14], a kind of FNO-based LOM.

**Experimental Verification.** We explore whether the performance degradation caused by truncating full-rank updates in the *parameter space* can be effectively mitigated by employing a lightweight, low-rank, and non-linear adapter that operates directly in the *representation space*. Specifically, we aim to determine if this adapter can accurately recover the functional shift that occurs due to such truncation. For more detailed experimental information, please refer to the Appendix C.9.

For the first Fourier- Attention block in DPOT we harvest the real Fourier activations $H \in \mathbb{R}^{N \times d}$ and their target outputs $Y = H\Delta W^\top$, where $\Delta W$ denotes the exact full-rank weight update. After a 90/10 % train–validation split we benchmark two surrogates: (i) a **two-layer MLP adapter** $f_{\mathrm{MLP}} : H \mapsto Y$ whose hidden width is $m$, and (ii) a **low-rank truncation** baseline obtained by replacing $\Delta W$ with its optimal rank-$r$ SVD approximation, which corresponds to an idealized LoRA module. Both models are evaluated by the root-mean-square error $\mathrm{RMSE} = \sqrt{|Y|^{-1}\|f(H) - Y\|_2^2}$ on the held-out set.

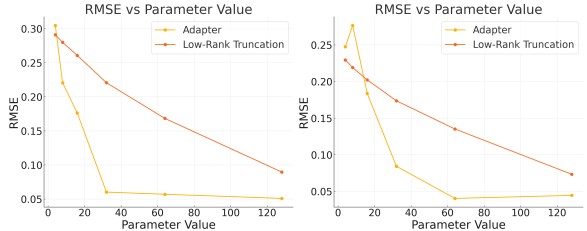

Figure 2: RMSE versus parameter (bottleneck dimension $m$ for Adapter and rank $r$ for Truncation) budget for the two–layer MLP adapter (yellow) and the low-rank truncation baseline (orange). Left: transonic dataset ($M = 1.0$). Right: low-Mach dataset ($M = 0.1$).

Figure 2 shows the RMSE achieved by a two-layer MLP adapter against the best rank-$r$ SVD truncation of $\Delta W$ (idealized LoRA) under identical parameter budgets and supplementary experimental results can be found at Appendix C.10. The advantage for Adapters is already noticeable at extremely small budgets ($m, r \leq 16$) and widens as capacity grows. In the transonic case ($M = 1.0$) the adapter with only $m = 64$ hidden units *halves* the error of a rank-128 truncation.

### 3.3 Spectral Energy Concentration in Low-Frequency Bands

To systematically gauge the relative importance of different spectral bands to predictive accuracy, we conduct a *spectral drop-high* experiment on a fully fine-tuned DPOT model evaluated on the three-dimensional Navier–Stokes benchmarks with random initial conditions at $M = 0.1$ and $M = 1.0$ and Turbulence initial conditions at $M = 1.0$.

A fixed mini-batch taken from the test set is transformed to Fourier space and its spectrum is uniformly partitioned into $N_b$ non-overlapping frequency bands. For each cut-off index $k \in \{0, \ldots, N_b - 1\}$ we zero out every coefficient that lies in a band whose index is $\geq k$, perform an inverse FFT,

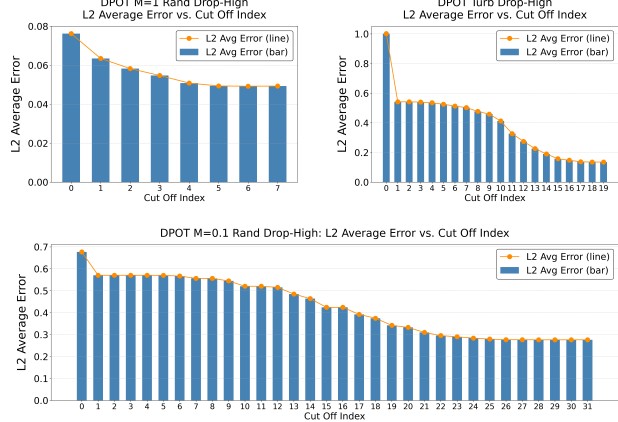

Figure 3: Mean $L_2$ error versus the cut-off band index $k$ for Rand $M = 1.0$ (upper left), Turb $M = 1.0$ (upper right), and Rand $M = 0.1$ (bottom).

and pass the filtered representation through the network. The mean $L_2$ relative error between the network output and the ground-truth target is then recorded. Sweeping $k$ from low to high therefore reveals how progressively discarding higher-frequency content influences the model's accuracy.

The resulting error curves for all datasets are presented in Figure 3. As the cut-off index $k$ increases, meaning progressively higher spectral bands are excised, the average $L_2$ error falls steeply at first and then flattens, eventually becoming almost insensitive to the removal of additional bands. The error curves suggest that most predictive power is captured once a relatively small set of low-frequency bands is retained; beyond this point, excising additional high-frequency bands yields only marginal further degradation. This behaviour is consistent with the spectral-truncation heuristics commonly used in FNO-style models [38, 54, 41].

To place the empirical finding on firmer ground, we extend classical harmonic analysis to time-continuous PDE solutions and, using Proposition 3.3, show that the cumulative energy of high-frequency modes decays polynomially with the cut-off radius, which implies that low-frequency modes carry the dominant share of the energy. The proof is provided in Appendix B.

**Proposition 3.3** (Quantitative Low–/High–Frequency Energy Split for PDE Solution). *Let $s > \frac{d}{2}$ and suppose $f \in C\big([0,T]; H^s(\mathbb{T}^d)\big)$, $\sup_{t \in [0,T]} \|f(t)\|_{H^s} \leq M$. Let $f(t,x) = \sum_{k \in \mathbb{Z}^d} \widehat{f}(t,k)\, e^{i\,k \cdot x}$, then for each $k \neq 0$,*

$$|\widehat{f}(t,k)| \leq M\,(1 + \|k\|^2)^{-s/2}, \tag{4}$$

*and for every integer $K \geq 1$ there exists $C = C(d,s)$ such that*

$$\sum_{\|k\| > K} |\widehat{f}(t,k)|^2 \leq C\,M^2\,K^{\,d-2s}, \qquad \sum_{\|k\| \leq K} |\widehat{f}(t,k)|^2 = \|f(t)\|_{L^2}^2 - O\big(K^{\,d-2s}\big). \tag{5}$$

Proposition 3.3 shows that, in Fourier space, PDE solutions concentrate most of their energy in relatively low-frequency modes while the high-frequency tail decays as $O(K^{d-2s})$.

Taken together, these findings indicate that low-frequency bands predominantly convey the global, energy-rich structure of fluid flows, arguing for higher-capacity adapters in that regime. Conversely, high-frequency bands are comparatively sparse and noise-prone; representing them with lightweight, low-rank transformations not only suffices for detail reconstruction but also serves as an effective spectral regularizer. Leveraging the complementary information carried by different frequency bands more effectively therefore represents a critical avenue for carrying out PEFT.

## 4  Methodology

For parameter-efficient fine-tuning, we retrofit each Fourier-domain mixing layer of the LOM with **Frequency-Adaptive Adapters (F-Adapters)**—bottleneck MLPs whose width varies per frequency band according to a governed formula (Eq. (6)). The design is *model-agnostic*: it can be plugged into any FFT-based layer without altering the host architecture's training recipe. The overall F-Adapter pipeline is illustrated in Figure 4.

**Fourier Representation.** Given an input tensor $\mathbf{x} \in \mathbb{R}^{B \times C \times H \times W \times L}$, we first perform a real 3-D FFT: $\hat{\mathbf{x}} = \mathrm{rFFTN}_{(2,3,4)}(\mathbf{x}) \in \mathbb{C}^{B \times M_1 \times M_2 \times M_3 \times C}$, where $M_i = \lfloor \frac{H}{2} \rfloor + 1$ for real transforms. The last dimension ($C$ channels) is split into $K$ non-overlapping *blocks* with equal width $d = C/K$.

**Band Partitioning.** We retain only the lowest $M = \min(M_1, M_2, M_3)$ spatial modes and partition them into $B$ contiguous *frequency bands* $0 = b_0 < b_1 < \cdots < b_B = M$, $\mathcal{B}_b = \{b_{b-1}, \ldots, b_b - 1\}$, so that every block–band slice can be processed independently. For most cases we set $B = 4$ and choose $b_b = \lfloor \frac{b}{B} M \rfloor$.

**Band-Specific Bottleneck Allocation.** Let $f_b = \frac{1}{2}(b_{b-1} + b_b)$ be the centre frequency of band $b$. We allocate a bottleneck width $r_b$ according to

$$r_b = \left\lfloor r_{\min} + (r_{\max} - r_{\min})\big(1 - \tfrac{f_b}{M}\big)^p \right\rfloor, \tag{6}$$

where $r_{\min}$, $r_{\max}$ and $p$ are hyper-parameters controlling the curvature. Lower bands receive wider $r_b$, while higher bands shrink toward $r_{\min}$.

**F-Adapter Micro-Architecture** In the DPOT-H [14] backbone, for each *block* $k \in [K]$ and *band* $b \in [B]$ in Fourier Attention Layer we attach three tiny adapters:

$$A_{k,b}^{\mathrm{in}} \colon \mathbb{R}^d \to \mathbb{R}^d, \quad A_{k,b}^{\mathrm{mid}} \colon \mathbb{R}^{d\,h_t} \to \mathbb{R}^{d\,h_t}, \quad A_{k,b}^{\mathrm{out}} \colon \mathbb{R}^d \to \mathbb{R}^d, \tag{7}$$

**Algorithm 1** F-Adapter PEFT Forward Pass in DPOT's [14] Fourier Attention Layer

$\hat{\mathbf{x}} \leftarrow \mathbf{rFFTN}(\mathbf{x})$      ▷ Step 1: FFT
$\hat{\mathbf{x}} \leftarrow \text{reshape}\big(\hat{\mathbf{x}}, (K, d), (M, M, M_t)\big)$      ▷ Step 2: reshape channels/modes
**for** $k \leftarrow 1$ **to** $K$ **do**      ▷ Step 3: band loop
    **for** $b \leftarrow 1$ **to** $B$ **do**
        $(i{:}j) \leftarrow \text{band\_indices}(b)$      ▷ 3-a: compute slice indices
        $\mathbf{z} \leftarrow \hat{\mathbf{x}}[:, i{:}j, i{:}j, 0{:}M_t, k]$      ▷ 3-b: extract complex slice
        $\mathbf{z_R} \leftarrow \Re(\mathbf{z})$ ; $\mathbf{z_I} \leftarrow \Im(\mathbf{z})$      ▷ 3-c: split real/imag
        $\mathbf{z_R} \leftarrow A^{\text{in}}_{k,b}(\mathbf{z_R})$ ; $\mathbf{z_I} \leftarrow A^{\text{in}}_{k,b}(\mathbf{z_I})$      ▷ 3-d: input adapter
        $(\mathbf{u_R}, \mathbf{u_I}) \leftarrow \text{fourier\_mix}(\mathbf{z_R}, \mathbf{z_I}, \mathbf{W}^{(1)}_k)$      ▷ 3-e: first Fourier mixing
        $\mathbf{u_R} \leftarrow \text{GELU}(\mathbf{u_R})$ ; $\mathbf{u_I} \leftarrow \text{GELU}(\mathbf{u_I})$      ▷ 3-f: activation
        $\mathbf{u_R} \leftarrow A^{\text{mid}}_{k,b}(\mathbf{u_R})$ ; $\mathbf{u_I} \leftarrow A^{\text{mid}}_{k,b}(\mathbf{u_I})$      ▷ 3-g: mid adapter
        $(\mathbf{v_R}, \mathbf{v_I}) \leftarrow \text{fourier\_mix}(\mathbf{u_R}, \mathbf{u_I}, \mathbf{W}^{(2)}_k)$      ▷ 3-h: second Fourier mixing
        $\mathbf{v_R} \leftarrow A^{\text{out}}_{k,b}(\mathbf{v_R})$ ; $\mathbf{v_I} \leftarrow A^{\text{out}}_{k,b}(\mathbf{v_I})$      ▷ 3-i: output adapter
        $\hat{\mathbf{x}}[:, i{:}j, i{:}j, 0{:}M_t, k] \leftarrow \mathbf{v_R} + i\,\mathbf{v_I}$      ▷ 3-j: scatter back
$\mathbf{x}' \leftarrow \text{iRFFTN}(\hat{\mathbf{x}})$      ▷ Step 4: IFFT
**return** $\mathbf{x}' + \mathbf{x}$      ▷ Step 5: residual

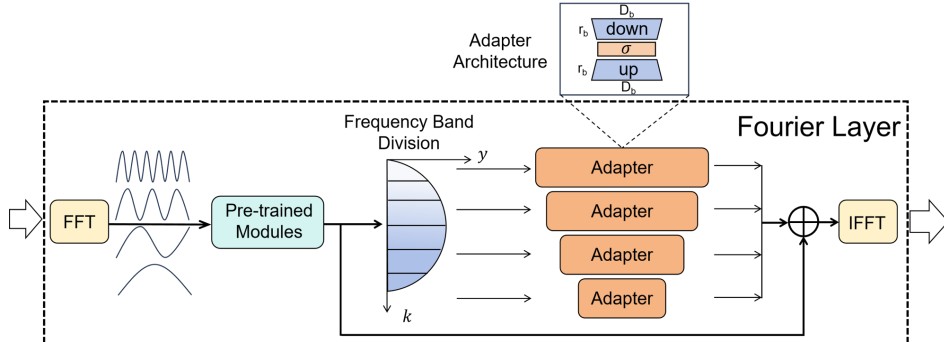

Figure 4: Pipeline for inserting Frequency-Adaptive Adapters (F-Adapters) between consecutive pre-trained Fourier sub-modules in a Fourier layer in LOMs.

where $h_t$ is the number of retained temporal modes. Each adapter implements the canonical bottleneck residuum

$$\mathbf{z}_{\text{down}} = \mathbf{W}^{\text{down}}_b \mathbf{z} + \mathbf{b}^{\text{down}}_b, \qquad\qquad \mathbf{z}_{\text{act}} = \sigma(\mathbf{z}_{\text{down}}),$$
$$\mathbf{z}_{\text{up}} = \mathbf{W}^{\text{up}}_b \mathbf{z}_{\text{act}} + \mathbf{b}^{\text{up}}_b, \qquad\qquad \tilde{\mathbf{z}} = \mathbf{z} + s_b\, \mathbf{z}_{\text{up}}, \tag{8}$$

with $\sigma = \text{GELU}$ and $s_b$ is a scalar. The matrices have shapes $\mathbf{W}^{\text{down}}_b \in \mathbb{R}^{r_b \times D}$, $\mathbf{W}^{\text{up}}_b \in \mathbb{R}^{D \times r_b}$, where $D \in \{d,\, d\,h_t\}$ depending on the adapter stage. The Parameter-Efficient Fine-Tuning forward pass of the F-Adapter within the DPOT backbone is outlined in Algorithm 1.

**Initialization and Training Details.** We adopt *zero-initialization* for every $\mathbf{W}^{\text{up}}_b$ and $\mathbf{b}^{\text{up}}_b$ so that, at the start of fine-tuning, the adapted path is an exact identity and does not perturb the pre-trained backbone. Down-projection weights are initialized with Kaiming-uniform initialization. All spectral kernels $\mathbf{W}^{(1)}, \mathbf{W}^{(2)}$ follow the scale definition $\frac{1}{d^2 h_t}$ from the Fourier Attention Later in DPOT.

**Computational Cost.** Let $N_s = BK$ denote the total number of block–band slices in one Fourier-Attention layer. The additional parameters introduced by the F-Adapters can be written as

$$\big|\Theta_{\text{F-Adapter}}\big| = \sum_{b=1}^{B} \Big(2\,d\,r_b + r_b\Big) \times K\,(2 + h_t) = (2d + 1)\,K\,(2 + h_t) \sum_{b=1}^{B} r_b. \tag{9}$$

Defining the average bottleneck width $\bar{r} = \frac{1}{B}\sum_{b=1}^{B} r_b$, this simplifies to

$$\big|\Theta_{\text{F-Adapter}}\big| = (2d + 1)\,B\,K\,(2 + h_t)\,\bar{r} = O\big(d\,B\,K\,h_t\,\bar{r}\big). \tag{10}$$

For typical settings, this overhead remains below 2% of the host model's total parameter count.

| Scheme | % Params | Rand | | | Turbulence | |
|---|---|---|---|---|---|---|
| | | Mem (GB) | L2RE ($M$=1.0) | L2RE ($M$=0.1) | Mem (GB) | L2RE |
| AdaLoRA [58] | 0.69% | 15.83 | 0.6726 | 0.6275 | 27.08 | 0.6795 |
| HydraLoRA [48] | 0.85% | 22.14 | 0.6333 | 0.6164 | 33.85 | 0.6888 |
| Prompt Tuning [20] | 1.03% | 19.82 | 0.6378 | 0.6127 | 23.37 | 0.6651 |
| Vanilla Adapter [17] | 1.16% | 15.85 | 0.5496 | 0.4893 | 25.41 | 0.4696 |
| FiLM Adapter [43] | 1.30% | 15.85 | 0.5655 | 0.5054 | 26.76 | 0.4987 |
| RandLoRA [1] | 1.36% | 15.86 | 0.6370 | 0.6125 | 24.69 | 0.6893 |
| LoRA [18] | 1.37% | 15.85 | 0.6395 | 0.6211 | 25.03 | 0.6842 |
| **F-Adapter (Ours)** | 1.91% | 15.88 | **0.5329** | **0.4639** | 26.90 | **0.4523** |
| SVFT [25] | 2.31% | 15.91 | 0.6375 | 0.5984 | 23.36 | 0.6655 |
| Full Fine-Tuning | 100.00% | 25.27 | 0.5391 | 0.4002 | 37.06 | 0.2382 |

Table 2: PEFT results on the 1B-parameter DPOT-H backbone for 3D Navier–Stokes forecasting.

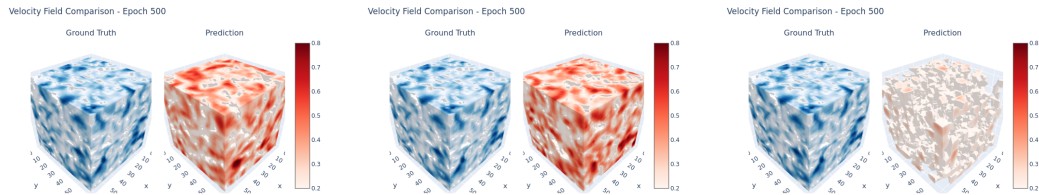

Figure 5: Side-by-side velocity field comparisons for Turbulence at epoch 500. From left to right: Vanilla Adapter, F-Adapter (Ours), and LoRA. Each panel shows ground-truth compared with prediction.

**Plug–and–Play Deployment.** Figure 4 depicts the *drop-in* procedure that enables an F-Adapter to be grafted onto **any** FFT-based Fourier layer appearing in FNO-style Large Operator Models (LOMs) without disturbing the host training recipe. First, the input field $\mathbf{x} \in \mathbb{R}^{B \times C \times H \times W \times L}$ is mapped to the spectral domain $\hat{\mathbf{x}} = \mathrm{rFFTN}(\mathbf{x})$. The resulting complex tensor is then passed through the *pre-trained and frozen* Fourier-mixing kernels of the backbone, upon which we retain the lowest $M$ spatial modes and slice them into $B$ contiguous radial frequency bands $\mathcal{B}_1, \ldots, \mathcal{B}_B$ according to $0 = b_0 < b_1 < \cdots < b_B = M$. For every block $k \in [K]$ and band $b \in [B]$ we attach a bottleneck MLP whose width $r_b$ is computed by Eq. (6). Each adapter performs the down–activation–up transformation defined in Eq. (8), writes the adapted coefficients back to their original spectral locations, and leaves all surrounding FFT logic untouched. Finally, the spectrum is converted back to physical space via $\mathbf{x}' = \mathrm{iRFFTN}(\hat{\mathbf{x}}_{\mathrm{adapted}})$ and added residually to the original signal. Because the procedure relies solely on the presence of an FFT/iFFT pair, it applies verbatim to every LOM that scales out of the FNO family. If a Fourier layer in an LOM contains *multiple* pre-trained sub-modules that require fine-tuning, one can interleave F-Adapters between those sub-modules following the workflow depicted in Figure 4. Algorithm 1 offers a representative instantiation of such a multi-module F-Adapter deployment.

## 5 Experiments

### 5.1 Main Experiments

Our principal comparison of the proposed F-Adapter against a suite of PEFT approaches widely used for LLMs follows the experimental protocol described in Section 3.1. In the 3D Navier–Stokes forecasting task in Table 2, our method lowers L2RE by from 16.7 % to 25.4 % while using nearly the same GPU memory as standard LoRA and its variants. Relative to the strongest baseline (Vanilla Adapter), F-Adapter improves accuracy by from 3.0 % to 5.2 % with a comparable parameter budget, which supports the value of frequency-adaptive capacity allocation. Side–by–side slice plots in Figure 5 of velocity magnitude show that F–Adapter retains filamentary vortical structures and reproduces both low– and high–energy regions with small local amplitude deviations. LoRA yields coarse block–like patches with muted intensities and loses fine–scale features, which is consistent with its collapsed spectrum.

Table 3 further reports results for PEFT methods on the 2D shallow-water equations (SWE-2D) and the 3D magnetohydrodynamics task (MHD-3D), two settings that demand high-frequency fidelity and broad spectral coverage. For MHD-3D, we follow the data processing protocol of Du et al. [9] but train on only 24 trajectories, creating a severe data-scarcity scenario. The results in Table 3 show that even under high-frequency regimes and limited data, F-Adapter maintains substantially higher accuracy, whereas alternative PEFT schemes struggle to adapt. These findings indicate that, instead of overlooking high-frequency content, F-Adapter partitions the spectrum so that all frequency bands are handled more efficiently and more effectively.

Full fine-tuning unsurprisingly achieves the best raw accuracy in most cases, but it requires ~50× more trainable parameters and ~1.4× the memory during training. Hence, F-Adapter strikes a favourable accuracy–efficiency trade-off and represents a practical alternative when computational resources or deployment budgets are constrained.

These findings substantiate our central claim: explicitly matching adapter capacity to the spectral characteristics of scientific operators is critical for effective and economical adaptation, whereas LoRA or prompt-based methods designed for language do not readily transfer to the SciML regime. For more Spectral analysis, please refer to Appendix C.12.

## 5.2 Ablation Studies

**Ablation on the Effect of the Dimension–Frequency Schedule.** To isolate the effect of the dimension–frequency schedule, we introduce an *F-Inverse-Adapter* for ablation study. In the F-Inverse-Adapter, we reverse the capacity allocation, giving higher-frequency bands larger bottleneck dimensions while matching F-Adapter's overall parameter count and memory consumption. Concretely, its bottleneck size is $\left\lfloor r_b = r_{\min} + (r_{\max} - r_{\min}) \left(\frac{f_b}{M}\right)^p \right\rfloor$, where $M$ is the total number of modes and $p$ is the hyperparameter exponent.

The F-Adapter, which assigns larger dimensions to low-frequency bands and smaller ones to high-frequency bands, outperforms both the vanilla Adapter and the reversed F-Inverse-Adapter in Table 5. This confirms that our dimension–frequency schedule is both reasonable and effective.

**Ablation on Different Types of Adapters for Fourier Domain.** We explore alternative strategies to address the challenges of applying Adapters in the Fourier domain. Motivated by Xiao et al. [52], which amortizes Fourier–kernel parameters with a KAN [30] to accommodate arbitrary high-frequency modes, we propose the *Fourier Adapter*. We substitute the adapter's down-projection and up-projection layers

| Scheme on DPOT | SWE-2D | | MHD-3D | |
|---|---|---|---|---|
| | L2RE | % Param | L2RE | % Param |
| AdaLoRA [58] | 0.1061 | 0.70% | 1.0022 | 0.69% |
| HydraLoRA [48] | 0.0956 | 0.88% | 0.9440 | 0.85% |
| Prompt Tuning [20] | 0.1050 | 0.11% | 0.9950 | 1.03% |
| Vanilla Adapter [17] | 0.0902 | 0.48% | 0.7226 | 1.16% |
| FiLM Adapter [43] | 0.0162 | 0.57% | 0.7593 | 1.30% |
| RandLoRA [1] | 0.1568 | 1.05% | 0.9800 | 1.36% |
| LoRA [18] | 0.1081 | 1.40% | 0.9845 | 1.37% |
| **F-Adapter (Ours)** | **0.0116** | 1.24% | **0.6341** | 1.91% |
| SVFT [25] | 0.0975 | 0.84% | 1.0004 | 2.31% |
| Full Fine-Tuning | 0.0023 | 100% | 0.4190 | 100% |

Table 3: PEFT results on the 1B-parameter DPOT-H backbone for 2D shallow-water equations (SWE-2D) and the 3D magnetohydrodynamic (MHD-3D) in data scarcity conditions.

with a FourierKAN layer [53], which parameterizes edge functions as truncated Fourier series, and integrate this module directly into the Fourier domain. Motivated by Zhang et al. [57], who applied Chebyshev polynomial bases within KANs to enhance PINNs for PDE solution, we substitute the adapter's linear layers with Chebyshev-based KAN modules to form the Chebyshev Adapter. Motivated by Tripura and Chakraborty [50], who incorporate wavelet transforms into operator learning, and by Zhao et al. [60], who employ the learnable wavelet-based activation function WaveAct for solving PDEs, we introduce the WaveAct Adapter—an adapter module that uses WaveAct as its nonlinear activation throughout the down- and up-projection layers. Architectural details for all the aforementioned Fourier-domain adapter improvements can be found in Appendix C.13.

Comparing the results in Table 4 with Table 2, we can conclude that most of the aforementioned adapters specifically designed for the Fourier domain do indeed outperform the Vanilla Adapter and LoRA. But *Chebyshev Adapter* slightly lags behind F-Adapter in accuracy and increases latency

| Scheme | % Params | Mem (GB) | FLOPs (G) | Time (ms) | L2RE ($M$=1) | L2RE ($M$=0.1) |
|---|---|---|---|---|---|---|
| **F-Adapter (Ours)** | 1.91% | 15.88 | 548.53 | **90.38** | **0.5329** | **0.4639** |
| Chebyshev Adapter | 2.18% | 16.19 | 554.80 | 268.02 | 0.5409 | 0.4757 |
| Fourier Adapter | 1.93% | 20.57 | 546.85 | 1449.54 | 0.6584 | 0.6053 |
| WaveAct Adapter | 1.16% | 15.85 | 547.47 | 92.69 | 0.5566 | 0.4691 |

Table 4: Computational cost and accuracy of different types of frequency-domain adapters on the 1B-parameter DPOT-H backbone.

by many times, reflecting the overhead of the dense Chebyshev polynomial expansion within the KAN architecture. The *Fourier Adapter*'s costly FourierKAN edge-function evaluations increase memory use by 29% and slow runtime tenfold, and its accuracy falls sharply, illustrating that naïvely adding high-order Fourier series exacerbates spectral aliasing. *WaveAct Adapter* equals F-Adapter in memory and nearly in speed, yet its accuracy lags, implying that learnable wavelet activations alone cannot fully capture the high-frequency dynamics of PDE solutions in Fourier domain.

This ablation study demonstrates that adaptively allocating the lightweight adapter's low-rank dimension according to spectral content (F-Adapter) is more effective than replacing the projection layers with heavier functional bases. Other ablation studies that assess how hyperparameter choices affect F-Adapter performance are reported in Appendix C.14.

| Scheme | Rand | Rand | Turb |
|---|---|---|---|
| F-Inverse-Adapter | 0.5664 | 0.4983 | 0.4747 |
| Vanilla Adapter [17] | 0.5496 | 0.4893 | 0.4696 |
| **F-Adapter (Ours)** | **0.5329** | **0.4639** | **0.4523** |

Table 5: Ablation on adapter dimension schedules. Columns report L2 relative error (L2RE) on the Rand dataset at $M = 1.0$ (first column), the Rand dataset at $M = 0.1$ (second column), and the Turb dataset at $M = 1.0$ (third column).

### 5.3 Discussions

In this section, we discuss the application of frequency-based capacity allocation to other non-FNO based LOMs. Although an FNO backbone provides direct access to frequency features, our main focus is on assigning each frequency band its own proper bottleneck dimension rather than strictly performing convolution in the frequency domain. This insight allows our F-Adapter to extend naturally to non-FFT architectures. On the pure transformer-based Poseidon [16] model, we estimate frequency energy for each Linear layer from adjacent-token differences, and for each Conv2d layer we perform a local real 2-D FFT on the convolution output to obtain an energy spectrum that guides the adapter's weight generation. The PEFT adapter itself still operates in the native spatial domain. Capacities are allocated to bands according to their energy following Equation (6), which equips the model with frequency awareness. Table 6 reports the resulting performance gains on SWE-2D.

We observe that adapters deliver strong results when the base model is an FNO, yet their effectiveness declines sharply on a transformer backbone. In contrast, LoRA and its variants demonstrate robust performance on transformer backbones, reflecting established best practices in fine-tuning LLMs. But our F-Adapter still narrows this gap by significantly improving adapter performance on transformers. Building on this insight, we introduce F-LoRA: it preserves the frequency-based capacity allocation of F-Adapter while replacing the bottleneck MLP with LoRA-style low-rank linear updates. For detailed design, please refer to Appendix C.15. F-LoRA achieves SOTA performance across a broad suite of PEFT methods in this setting.

| Scheme on Poseidon | L2RE | % Param |
|---|---|---|
| Prompt Tuning [20] | $> 1.0$ | 0.07% |
| LoRA [18] | 0.4010 | 2.07% |
| RandLoRA [1] | 0.3134 | 2.07% |
| Vanilla Adapter [17] | 0.6231 | 2.18% |
| AdaLoRA [58] | 0.3756 | 2.32% |
| HydraLoRA [48] | 0.3474 | 2.57% |
| FiLM Adapter [43] | 0.4567 | 3.19% |
| SVFT [25] | 0.6742 | 4.22% |
| F-Adapter (Ours) | 0.4311 | 4.17% |
| **F-LoRA (Ours)** | **0.2746** | 4.78% |
| Full Fine-Tuning | 0.1534 | 100% |

Table 6: PEFT results on the Poseidon backbone for 2D shallow-water equations (SWE-2D).

## 6 Conclusion

We provide the first systematic PEFT study for pretrained LOMs, exposing a depth-amplified spectral error floor in LoRA. We prove that adapters avoid this limit and, guided by Fourier energy analysis, design F-Adapters that match capacity to modal energy. Updating at most 2% of weights, they set SOTA records in L2REs on several challenging partial differential equation tasks, validating a principled and efficient route for fine-tuning LOMs.

## Acknowledgements

We would like to thank the anonymous reviewers and area chairs for their helpful comments. This work is supported by NSFC 62306252, Hong Kong ECS award 27309624, Guangdong NSF 2024A1515012444, and the central fund from HKU IDS.

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

# A  Preliminary

Let $\mathbf{h} \in \mathbb{R}^d$ denote the hidden activation of a transformer sub-layer and let $\mathbf{W}_0 \in \mathbb{R}^{d \times d}$ be the frozen projection learned during pre-training.

**Low-Rank Adaptation (LoRA)**    LoRA [18] injects a rank-$r$ correction into $\mathbf{W}_0$ while keeping it frozen:

$$\mathbf{h}_{\text{out}} = \big(\mathbf{W}_0 + \Delta\mathbf{W}\big)\mathbf{h}, \quad \Delta\mathbf{W} = \alpha\, \mathbf{BA}, \tag{11}$$

$$\mathbf{A} \in \mathbb{R}^{r \times d}, \quad \mathbf{B} \in \mathbb{R}^{d \times r}, \quad \alpha = \tfrac{\lambda}{r}, \tag{12}$$

so that only $\mathbf{A}$ and $\mathbf{B}$—totalling $2rd$ parameters—are updated.

**Bottleneck Adapter**    Adapters [17] append a two-layer bottleneck MLP with a residual gate $s$:

$$\begin{aligned} \mathbf{h}_{\text{down}} &= \mathbf{W}_{\text{down}}\mathbf{h} + \mathbf{b}_{\text{down}}, & \mathbf{h}_{\text{act}} &= f(\mathbf{h}_{\text{down}}), \\ \mathbf{h}_{\text{up}} &= \mathbf{W}_{\text{up}}\mathbf{h}_{\text{act}} + \mathbf{b}_{\text{up}}, & \mathbf{h}_{\text{out}} &= \mathbf{h} + s\,\mathbf{h}_{\text{up}}, \end{aligned} \tag{13}$$

where $\mathbf{W}_{\text{down}} \in \mathbb{R}^{r \times d}$, $\mathbf{W}_{\text{up}} \in \mathbb{R}^{d \times r}$, $f(\cdot) = \text{GELU}$, $r \ll d$, $s \in \mathbb{R}_{>0}$. The adapter adds $2rd + d + r$ trainable parameters and reduces to the identity when $s = 0$.

Both LoRA and adapters thus enable parameter-efficient fine-tuning by confining learning to small, task-specific subspaces while preserving the frozen pre-trained backbone.

# B  Mathematical Proofs

**Lemma 1** (LoRA error lower bound). *Let $\Delta W \in \mathbb{R}^{d \times d}$ admit the singular-value decomposition $\Delta W = U\Sigma V^\top$ with singular values $\sigma_1 \geq \sigma_2 \geq \cdots \geq \sigma_d \geq 0$ and orthonormal singular vectors $U = [u_1, \ldots, u_d]$, $V = [v_1, \ldots, v_d]$. For any factorization $B \in \mathbb{R}^{d \times r}$, $A \in \mathbb{R}^{r \times d}$ and any $x \in \mathbb{R}^d$,*

$$\big\|\Delta W\, x - BAx\big\| \geq \sqrt{\sum_{i=r+1}^{d} \sigma_i^2 (v_i^\top x)^2}. \tag{14}$$

*Furthermore, for the worst-case lower bound of the LoRA error we obtain*

$$\sup_{\|x\|_2 = 1} \big\|\Delta W\, x - BA\, x\big\| \geq \sigma_{r+1}. \tag{15}$$

*Proof.* Because $\Delta W = U\Sigma V^\top$, every $x \in \mathbb{R}^d$ satisfies

$$\Delta W x = U\Sigma V^\top x = \sum_{i=1}^{d} \sigma_i \,(v_i^\top x)\, u_i. \tag{16}$$

By the orthogonality of $V = [v_1, \ldots, v_d]$, one has $V^\top V = I$, hence

$$x = I\, x = (V^\top V)\, x = V^\top (Vx) = \sum_{i=1}^{d} (v_i^\top x)\, v_i. \tag{17}$$

Since $\text{rank}(BA) \leq r$, $\text{im}(BA) \subseteq \text{span}\{u_1, \ldots, u_r\}$, so there exist scalars $\alpha_1, \ldots, \alpha_r$ with

$$BAx = \sum_{i=1}^{r} \alpha_i \, u_i. \tag{18}$$

Define the error $e(x) = \Delta W x - BAx$. Substituting (16) and (18) gives

$$e(x) = \sum_{i=1}^{r} \big[\sigma_i(v_i^\top x) - \alpha_i\big] u_i + \sum_{i=r+1}^{d} \sigma_i\,(v_i^\top x)\, u_i. \tag{19}$$

The orthonormality of $\{u_i\}$ implies

$$\|e(x)\|^2 = \sum_{i=1}^{r} \left[\sigma_i(v_i^\top x) - \alpha_i\right]^2 + \sum_{i=r+1}^{d} \sigma_i^2(v_i^\top x)^2. \tag{20}$$

Because the first sum can be made arbitrarily small by a suitable choice of $\alpha_i$, the second sum furnishes an unavoidable contribution:

$$\|\Delta W x - BAx\| \geq \sqrt{\sum_{i=r+1}^{d} \sigma_i^2(v_i^\top x)^2}. \tag{21}$$

It completes the proof of LoRA error's lower bound.

Next, we present the proof establishing the worst-case lower bound for the LoRA error.

Let $S := \text{im}(BA) \subseteq \mathbb{R}^d$ be the column space of $BA$. Since $\text{rank}(BA) \leq r$, we have:

$$\dim(S) \leq r \quad \text{and} \quad \dim(S^\perp) \geq d - r. \tag{22}$$

Consider the $(r+1)$-dimensional subspace spanned by the first $r+1$ right singular vectors:

$$\mathcal{V}_{r+1} := \text{span}\{v_1, \ldots, v_{r+1}\}. \tag{23}$$

By the subspace intersection theorem:

$$\dim(S^\perp \cap \mathcal{V}_{r+1}) \geq \dim(\mathcal{V}_{r+1}) + \dim(S^\perp) - d \geq (r+1) + (d-r) - d = 1. \tag{24}$$

Thus, there exists a unit vector $x_0 \in S^\perp \cap \mathcal{V}_{r+1}$.

Decompose $x_0$ in the singular vector basis:

$$x_0 = \sum_{i=1}^{r+1} \alpha_i v_i \quad \text{with} \quad \sum_{i=1}^{r+1} \alpha_i^2 = 1. \tag{25}$$

The approximation error satisfies:

$$\|\Delta W x_0 - BAx_0\|^2 = \|P_{S^\perp}(\Delta W x_0)\|^2 + \|P_S(\Delta W x_0) - BAx_0\|^2 \tag{26}$$

$$\geq \|P_{S^\perp}(\Delta W x_0)\|^2. \tag{27}$$

Using the SVD of $\Delta W$:

$$\Delta W x_0 = \sum_{i=1}^{r+1} \sigma_i \alpha_i u_i. \tag{28}$$

Since $x_0 \in S^\perp$ and $BAx_0 \in S$, we have $P_{S^\perp}(BAx_0) = 0$, thus:

$$P_{S^\perp}(\Delta W x_0) = \sum_{i=1}^{r+1} \sigma_i \alpha_i P_{S^\perp} u_i. \tag{29}$$

From the optimality of $\Delta W_r$, for any unit $y \in \mathbb{R}^d$:

$$\|\Delta W y - \Delta W_r y\| \geq \sigma_{r+1}. \tag{30}$$

Specifically for $y = x_0 \in \mathcal{V}_{r+1}$:

$$\|\Delta W x_0\|^2 = \sum_{i=1}^{r+1} \sigma_i^2 \alpha_i^2 \geq \sigma_{r+1}^2. \tag{31}$$

Combining (27) and (31):

$$\|\Delta W x_0 - BAx_0\|^2 \geq \|P_{S^\perp}(\Delta W x_0)\|^2 \tag{32}$$

$$\geq \|\Delta W x_0\|^2 - \|P_S(\Delta W x_0)\|^2 \tag{33}$$

$$\geq \sigma_{r+1}^2 - \sum_{i=1}^{r} \sigma_i^2 \alpha_i^2 \tag{34}$$

$$\geq \sigma_{r+1}^2 - \sigma_1^2 \sum_{i=1}^{r} \alpha_i^2 \tag{35}$$

$$\geq \sigma_{r+1}^2 - \sigma_1^2(1 - \alpha_{r+1}^2). \tag{36}$$

The maximum is achieved when $\alpha_{r+1} = 1$, giving:

$$\|\Delta W x_0 - BAx_0\| \geq \sigma_{r+1}. \tag{37}$$

Since there exists at least one $x_0$ for which Eq. 37 holds, the supremum over all $x \neq 0$ satisfies:

$$\sup_{x \neq 0} \|\Delta W x - BAx\| \geq \sigma_{r+1}. \tag{38}$$

$\square$

**Remark 1.** *The worst-Case lower bound coincides with the optimal spectral-norm error $\|\Delta W - \Delta W_r\|_2 = \sigma_{r+1}$ given by the classical Eckart–Young–Mirsky theorem[10], and is therefore tight.*

**Remark 2.** *Lemma 1 demonstrates that LoRA cannot attain zero approximation error; while its worst-case error is governed by the $(r+1)$-st singular value, LoRA yields meaningful improvement only when this singular value is sufficiently small.*

**Proposition 3.1** ( Block-wise LoRA lower bound ). *Let $\Delta W_g = \text{blockdiag}\big(\Delta W^{(1)}, \dots, \Delta W^{(K)}\big)$ be the block-wise model parameter updates and $BA = \text{blockdiag}\big(B^{(1)} A^{(1)}, \dots, B^{(K)} A^{(K)}\big)$ be the block-wise low-rank approximation, where $B^{(k)} \in \mathbb{C}^{d \times r}, A^{(k)} \in \mathbb{C}^{r \times d}$. Then, for any input $x$, the approximation error for block-wise LoRA satisfies*

$$\big\|(\Delta W_g - BA)x\big\| \geq \left(\sum_{k=1}^{K} \sum_{i=r+1}^{d} \sigma_{k,i}^2 \,(v_{k,i}^\top x_k)^2\right)^{1/2}. \tag{1}$$

*In particular, the worst-case operator-norm error obeys*

$$\sup_{\|\hat{x}\|_2 = 1} \big\|(\Delta W_g - BA)\hat{x}\big\|_2 \geq \sigma_{Kr+1}(\Delta W_g) \tag{2}$$

*Proof.* Since $V_k = [v_{k,1}, \dots, v_{k,d}] \in \mathbb{R}^{d \times d}$ is orthogonal,

$$V_k V_k^\top = I_d. \tag{39}$$

For any $x_k \in \mathbb{R}^d$,

$$x_k = V_k\big(V_k^\top x_k\big) = \sum_{i=1}^{d} (v_{k,i}^\top x_k)\, v_{k,i}. \tag{40}$$

Set

$$e_k := \Delta W_k x_k - B_k A_k x_k. \tag{41}$$

With $\Delta W_k = U_k \Sigma_k V_k^\top$ and $\text{im}(B_k A_k) \subseteq \text{span}\{u_{k,1}, \dots, u_{k,r}\}$, there exist scalars $\alpha_{k,i}$ $(1 \leq i \leq r)$ such that

$$\Delta W_k x_k = \sum_{i=1}^{d} \sigma_{k,i}(v_{k,i}^\top x_k)\, u_{k,i}, \tag{42}$$

$$B_k A_k x_k = \sum_{i=1}^{r} \alpha_{k,i}\, u_{k,i}. \tag{43}$$

Hence

$$e_k = \sum_{i=1}^{r} \big[\sigma_{k,i}(v_{k,i}^\top x_k) - \alpha_{k,i}\big] u_{k,i} + \sum_{i=r+1}^{d} \sigma_{k,i}(v_{k,i}^\top x_k)\, u_{k,i}. \tag{44}$$

Orthonormality of $\{u_{k,i}\}$ implies

$$\|e_k\|^2 = \sum_{i=1}^{r} \big[\sigma_{k,i}(v_{k,i}^\top x_k) - \alpha_{k,i}\big]^2 + \sum_{i=r+1}^{d} \sigma_{k,i}^2(v_{k,i}^\top x_k)^2 \geq \sum_{i=r+1}^{d} \sigma_{k,i}^2(v_{k,i}^\top x_k)^2. \tag{45}$$

Since $(\Delta W_g - BA)x = (e_1, \ldots, e_K)^\top$ and the blocks are orthogonal,

$$\|(\Delta W_g - BA)x\|^2 = \sum_{k=1}^{K} \|e_k\|^2 \geq \sum_{k=1}^{K} \sum_{i=r+1}^{d} \sigma_{k,i}^2 (v_{k,i}^\top x_k)^2, \tag{46}$$

and taking square roots of (46) yields the stated bound.

Next, we present the proof establishing the worst-case lower bound for the block-wise LoRA error. Define

$$\|\Delta W_g - BA\|_{\text{op}} = \sup_{\|\hat{x}\|_2 = 1} \|(\Delta W_g - BA)\hat{x}\|_2. \tag{47}$$

For any unit vector $\hat{x} = (x_1, \ldots, x_K)^\top$,

$$(\Delta W_g - BA)\hat{x} = (D_1 x_1, \ldots, D_K x_K)^\top, \quad \|(\Delta W_g - BA)\hat{x}\|_2^2 = \sum_{k=1}^{K} \|D_k x_k\|_2^2. \tag{48}$$

Writing the SVD $\Delta W_k = U_k \Sigma_k V_k^\top$ with $\Sigma_k = \text{diag}(\sigma_{k,1}, \ldots, \sigma_{k,d})$, any $x \in \mathbb{R}^d$ expands as $x = \sum_i (v_{k,i}^\top x) v_{k,i}$ and

$$D_k x = \sum_{i=1}^{r} \left[ \sigma_{k,i}(v_{k,i}^\top x) - \alpha_{k,i} \right] u_{k,i} + \sum_{i=r+1}^{d} \sigma_{k,i}(v_{k,i}^\top x)\, u_{k,i},$$

for some $\alpha_{k,1}, \ldots, \alpha_{k,r}$. Choosing $x = v_{k,r+1}$ gives

$$\|D_k\|_{\text{op}} \geq \|D_k v_{k,r+1}\| = \sigma_{k,r+1}. \tag{49}$$

Taking the maximum over $k$ and using (47) yields (2). Finally, since $\Delta W_g$ is block-diagonal its singular values are the multiset $\{\sigma_{k,i}\}$, so (2) holds and the proof is complete.

$\square$

**Lemma 2** (Universal Approximation Theorem for Adapters). *Let $K \subset \mathbb{R}^D$ be compact, $\sigma \colon \mathbb{R} \to \mathbb{R}$ a continuous non-affine function, and $f \in C(K; \mathbb{R}^D)$. For every $\varepsilon > 0$, there exist parameters $m \in \mathbb{N}$, $V \in \mathbb{R}^{m \times D}$, $U \in \mathbb{R}^{D \times m}$, $b \in \mathbb{R}^m$, and $c \in \mathbb{R}^D$ such that*

$$h(x) = U\,\sigma(Vx + b) + c \quad \text{satisfies} \quad \sup_{x \in K} \|h(x) - f(x)\|_\infty < \varepsilon. \tag{50}$$

*Proof.* Define the hypothesis classes:

$$\mathcal{H}_0 = \left\{ \sum_{i=1}^{m} u_i \sigma(w_i^\top x + b_i) \;\middle|\; u_i \in \mathbb{R}^D, w_i \in \mathbb{R}^D, b_i \in \mathbb{R} \right\}, \quad \mathcal{H} = \mathcal{H}_0 + \{c\}. \tag{51}$$

*Step 1: Density Argument by Contradiction.* Assume $\overline{\mathcal{H}} \neq C(K; \mathbb{R}^D)$. By Hahn-Banach theorem, there exists a non-zero continuous linear functional $L \in (C(K; \mathbb{R}^D))^*$ such that:

$$L(h) = 0 \quad \forall h \in \mathcal{H}. \tag{52}$$

*Step 2: Riesz-Markov-Kakutani Representation.* For vector-valued continuous functions, there exists a $\mathbb{R}^D$-valued Radon measure $\boldsymbol{\mu} = (\mu_1, \ldots, \mu_D)$ such that:

$$L(g) = \sum_{j=1}^{D} \int_K g_j(x) d\mu_j(x), \quad \forall g \in C(K; \mathbb{R}^D). \tag{53}$$

The annihilation condition becomes:

$$\sum_{j=1}^{D} \int_K h_j(x) d\mu_j(x) = 0 \quad \forall h \in \mathcal{H}. \tag{54}$$

*Step 3: Constant Function Elimination.* Testing with constant functions $h(x) \equiv c_0 \in \mathbb{R}^D$:

$$\sum_{j=1}^{D} c_{0,j} \mu_j(K) = 0 \quad \forall c_0 \in \mathbb{R}^D. \tag{55}$$

This implies the total mass vanishes for each component:

$$\mu_j(K) = 0, \quad \forall 1 \leq j \leq D. \tag{56}$$

*Step 4: Single Neuron Analysis.* For any direction $w \in \mathbb{R}^D$, bias $b \in \mathbb{R}$, and basis vector $e_j$, consider:

$$h(x) = e_j \sigma(w^\top x + b) \in \mathcal{H}_0. \tag{57}$$

Substitution into (54) gives:

$$\int_K \sigma(w^\top x + b) d\mu_j(x) = 0 \quad \forall w \in \mathbb{R}^D, b \in \mathbb{R}, 1 \leq j \leq D. \tag{58}$$

*Step 5: Projection to 1D Measures.* For each $w \in \mathbb{R}^D$, define projected measures $\nu_{j,w}$ on $\mathbb{R}$ by:

$$\nu_{j,w}(A) := \mu_j \left( \{x \in K \,|\, w^\top x \in A\} \right) \quad \text{for Borel sets } A \subseteq \mathbb{R}. \tag{59}$$

Equation (58) becomes:

$$\int_{\mathbb{R}} \sigma(t + b) d\nu_{j,w}(t) = 0 \quad \forall b \in \mathbb{R}. \tag{60}$$

*Step 6: Fourier Analytic Argument.* Let $\mathcal{F}$ denote the Fourier transform. For tempered distributions:

$$\mathcal{F}[\sigma * \nu_{j,w}](\omega) = \mathcal{F}[\sigma](\omega) \cdot \mathcal{F}[\nu_{j,w}](\omega) \tag{61}$$

$$= \widehat{\sigma}(\omega) \cdot \widehat{\nu_{j,w}}(\omega) = 0 \quad \forall \omega \in \mathbb{R}. \tag{62}$$

*Lemma 6.1 (Non-vanishing spectrum).* For non-affine $\sigma \in C(\mathbb{R}) \setminus \mathcal{P}_1$, $\widehat{\sigma}$ is not identically zero. Specifically:

- If $\sigma$ is sigmoidal: $\widehat{\sigma}(\omega)$ has exponential decay but $\text{supp}(\widehat{\sigma}) = \mathbb{R}$

- For ReLU: $\widehat{\sigma}(\omega) = \pi \delta(\omega) + \frac{1}{i\omega}$ (in distribution sense)

- GeLU: $\widehat{\text{GeLU}}(\omega)$ is analytic and non-zero on $\mathbb{R} \setminus \{0\}$

Thus $\widehat{\nu_{j,w}} \equiv 0$ in (62), implying $\nu_{j,w} \equiv 0$.

*Step 7: Cramér-Wold Device.* For any $w \in \mathbb{R}^D$, the projected measure satisfies:

$$\nu_{j,w}(A) = \mu_j \left( x \in K \,|\, w^\top x \in A \right) = 0 \quad \forall A \subseteq \mathbb{R}. \tag{63}$$

By Cramér-Wold theorem [8], this implies:

$$\mu_j(B) = 0 \quad \forall \text{Borel } B \subseteq K, \ 1 \leq j \leq D. \tag{64}$$

Contradicting $L \neq 0$ in (53). Therefore $\overline{\mathcal{H}} = C(K; \mathbb{R}^D)$.

*Step 8: Approximation Construction.* Given $f \in C(K; \mathbb{R}^D)$ and $\varepsilon > 0$, by density there exists $h \in \mathcal{H}$ with:

$$\|h - f\|_{C(K)} = \sup_{x \in K} \max_{1 \leq j \leq D} |h_j(x) - f_j(x)| < \varepsilon. \tag{65}$$

This completes the universal approximation property. $\qquad\square$

**Remark 3** (Measure-Theoretic Details). *All measures are Radon measures by the Riesz-Markov-Kakutani theorem. Fourier transforms are interpreted in the distributional sense. The Cramér-Wold theorem applies to finite Borel measures*

**Remark 4** (Activation Function Spectrum). *The critical requirement is $\widehat{\sigma} \not\equiv 0$, satisfied by:*

- *Non-polynomial analytic functions: $\sigma(t) = e^t/(1 + e^t)$*

- *Piecewise linear functions with $\sigma'' \neq 0$ distributionally*

- *Functions with non-vanishing generalized spectrum*

**Remark 5.** *By contrast to Lemma 1, Lemma 2 proves a universal approximation theorem for adapters: the presence of nonlinear activation endows them with strictly greater expressive power than LoRA.*

**Lemma 3** (Spectral decay from Sobolev regularity). *Let $g\colon \mathbb{T}^d \to \mathbb{C}$ be $2\pi$-periodic with all weak derivatives up to order $\alpha \in \mathbb{N}$ in $L^1(\mathbb{T}^d)$, where $\alpha > d/2$. Denote its Fourier coefficients by*

$$g_k = \frac{1}{(2\pi)^d} \int_{\mathbb{T}^d} g(x) e^{-ik\cdot x}\, dx, \quad k \in \mathbb{Z}^d.$$

*Then there exists $C > 0$ such that*

$$|g_k| \leq C(1 + \|k\|^2)^{-\alpha/2}, \quad \forall k \in \mathbb{Z}^d.$$

*Proof.* Let $\alpha \in \mathbb{N}$ with $\alpha > d/2$ and fix a multi-index $m = (m_1, \ldots, m_d)$ satisfying $|m| = \sum_{j=1}^d m_j = \alpha$. For any test function $\phi \in C^\infty(\mathbb{T}^d)$, integration by parts in the distributional sense yields

$$\int_{\mathbb{T}^d} \partial^m g(x)\phi(x)dx = (-1)^{|m|} \int_{\mathbb{T}^d} g(x)\partial^m \phi(x)dx. \tag{66}$$

Applying this to $\phi(x) = e^{-ik\cdot x}$ and noting that $\partial^m e^{-ik\cdot x} = (-ik)^m e^{-ik\cdot x}$, we derive

$$\int_{\mathbb{T}^d} g(x)e^{-ik\cdot x}dx = \frac{1}{(-ik)^m} \int_{\mathbb{T}^d} \partial^m g(x)e^{-ik\cdot x}dx. \tag{67}$$

Consequently, the Fourier coefficients satisfy

$$g_k = \frac{1}{(2\pi)^d} \frac{1}{(-ik)^m} \int_{\mathbb{T}^d} \partial^m g(x)e^{-ik\cdot x}dx. \tag{68}$$

Taking absolute values and applying Hölder's inequality, we obtain

$$|g_k| \leq \frac{1}{(2\pi)^d|k^m|} \int_{\mathbb{T}^d} |\partial^m g(x)|dx = \frac{\|\partial^m g\|_{L^1}}{(2\pi)^d|k^m|}. \tag{69}$$

To bound $|k^m| = \prod_{j=1}^d |k_j|^{m_j}$, observe that by the arithmetic-geometric mean inequality,

$$\prod_{j=1}^d |k_j|^{m_j} \geq \left(\frac{\|k\|}{\sqrt{d}}\right)^\alpha, \tag{70}$$

where $\|k\| = \sqrt{k_1^2 + \cdots + k_d^2}$. Substituting this into the estimate for $|g_k|$ gives

$$|g_k| \leq \frac{d^{\alpha/2}\|\partial^m g\|_{L^1}}{(2\pi)^d}\|k\|^{-\alpha}. \tag{71}$$

For low-frequency modes with $\|k\| < 1$, the bound $|g_k| \leq \|g\|_{L^1}$ holds trivially. Combining both cases by defining

$$C = \max\left(\sup_{\|k\|<1} |g_k|, \frac{d^{\alpha/2}\|\partial^m g\|_{L^1}}{(2\pi)^d}\right), \tag{72}$$

we achieve the unified decay estimate

$$|g_k| \leq C(1 + \|k\|^2)^{-\alpha/2}, \quad \forall k \in \mathbb{Z}^d. \tag{73}$$

Sharpness follows by considering test functions $g(x) = \prod_{j=1}^d (1 - \cos x_j)^\beta$ with $\beta > \alpha$, where direct calculation shows $|g_k| \asymp \|k\|^{-2\beta}$. $\qquad\square$

**Lemma 4** (Spatial-domain adapter error bound). *Let a single–layer adapter in the spatial domain produce perturbations $\delta_i$ at grid points $i \in I \subseteq \{1, \ldots, N\}^d$ with $|\delta_i| \leq \varepsilon_m$ uniformly, and suppose $\#I = K^d$. Then the global $\ell^2$–error*

$$e \in \mathbb{R}^{N^d}, \quad e_i = \begin{cases} \delta_i, & i \in I, \\ 0, & i \notin I, \end{cases}$$

*satisfies*

$$\|e\| = \left(\sum_{i \in I} |\delta_i|^2\right)^{1/2} \leq \sqrt{K^d}\, \varepsilon_m = O\big(K^{d/2}\big), \tag{74}$$

*so in the absence of any decay in the perturbations one only obtains the spatial-domain rate $O(K^{d/2})$.*

*Proof.* By definition,

$$\|e\|^2 = \sum_{i \in I} |\delta_i|^2 \leq \sum_{i \in I} \varepsilon_m^2 = K^d\, \varepsilon_m^2,$$

and taking square roots yields $\|e\| \leq K^{d/2}\, \varepsilon_m = O(K^{d/2})$. $\qquad\square$

**Proposition 3.2** (Frequency-selective approximation of adapters). *Let $|g_k| \leq C\langle k\rangle^{-\alpha}$ with $\alpha > \frac{d}{2}$. For any $\varepsilon > 0$ there exist frequency truncation radius $K > 0$ and adapter bottleneck width $m \in \mathbb{N}$ such that the Fourier-domain adapter $\widehat{g}$ obeys*

$$\|e\| := \|\mathcal{F}^{-1}(g) - \mathcal{F}^{-1}(\widehat{g})\|_2 < \varepsilon, \qquad \|e\| = O\big(K^{\frac{d}{2}-\alpha}\big) + O\big(K^{\frac{d}{2}} e^{-cm}\big). \tag{3}$$

*Proof.* Let

$$e = \mathcal{F}^{-1}\big(g(\hat{x}) - \widehat{g}(\hat{x})\big). \tag{75}$$

By the unitarity of $\mathcal{F}^{-1}$ we have

$$\|e\|^2 = \sum_{k=1}^{N} \big|g_k(\hat{x}_k) - \widehat{g}_k(\hat{x}_k)\big|^2. \tag{76}$$

Split the sum in (76) into the high-frequency part $\langle k\rangle > K$ and the low-frequency part $\langle k\rangle \leq K$. For the high-frequency truncation we use $|\widehat{g}_k| = 0$ and the decay hypothesis $|g_k(\hat{x}_k)| \leq C\langle k\rangle^{-\alpha}$:

$$\sum_{\langle k\rangle > K} \big|g_k(\hat{x}_k)\big|^2 \leq C^2 \sum_{\langle k\rangle > K} \langle k\rangle^{-2\alpha} \leq C^2 \int_K^\infty \frac{r^{d-1}}{(1+r^2)^\alpha}\, dr. \tag{77}$$

Setting $r = \sqrt{s}$ with $dr = \frac{1}{2\sqrt{s}}ds$ gives

$$\int_K^\infty \frac{r^{d-1}}{(1+r^2)^\alpha}\, dr = \frac{1}{2}\int_{K^2}^\infty \frac{s^{\frac{d-2}{2}}}{(1+s)^\alpha}\, ds \leq \frac{1}{2} C_\alpha \int_{K^2}^\infty s^{\frac{d}{2}-\alpha-1}\, ds = \frac{C_\alpha}{2(\alpha - \frac{d}{2})} K^{d-2\alpha}, \tag{78}$$

so that

$$\sum_{\langle k\rangle > K} \big|g_k(\hat{x}_k)\big|^2 = O\big(K^{d-2\alpha}\big). \tag{79}$$

Taking square-roots yields the high-frequency contribution

$$\left(\sum_{\langle k\rangle > K} \big|g_k - \widehat{g}_k\big|^2\right)^{1/2} = A\, K^{\frac{d}{2}-\alpha}, \quad A = \sqrt{\frac{C^2\, C_\alpha}{2(\alpha - \frac{d}{2})}}. \tag{80}$$

For the low-frequency part $\langle k\rangle \leq K$, the adapter achieves exponential uniform accuracy:

$$\big|g_k(\hat{x}_k) - \widehat{g}_k(\hat{x}_k)\big| \leq \varepsilon_m\, e^{-cm}, \tag{81}$$

hence

$$\sum_{\langle k\rangle \leq K} \big|g_k - \widehat{g}_k\big|^2 \leq K^d\, \varepsilon_m^2\, e^{-2cm}, \tag{82}$$

and after taking square-roots the low-frequency contribution is

$$\left(\sum_{\langle k\rangle \leq K} \big|g_k - \widehat{g}_k\big|^2\right)^{1/2} = B\, K^{\frac{d}{2}} e^{-cm}, \quad B = \varepsilon_m. \tag{83}$$

Combining these two estimates with (76) gives

$$\|e\| \leq A\,K^{\frac{d}{2}-\alpha} + B\,K^{\frac{d}{2}}e^{-cm}. \tag{84}$$

To ensure $\|e\| < \varepsilon$, it suffices to choose $K$ and $m$ such that

$$A\,K^{\frac{d}{2}-\alpha} < \frac{\varepsilon}{2} \quad \Longrightarrow \quad K > \left(\frac{2A}{\varepsilon}\right)^{1/(\alpha-\frac{d}{2})}, \tag{85}$$

$$B\,K^{\frac{d}{2}}e^{-cm} < \frac{\varepsilon}{2} \quad \Longrightarrow \quad m > \frac{1}{c}\ln\!\left(\frac{2B\,K^{\frac{d}{2}}}{\varepsilon}\right). \tag{86}$$

Since $\alpha > d/2$, the exponent $1/(\alpha - \frac{d}{2})$ is positive, and thus one can always pick finite $K$ and then $m$ to satisfy both inequalities. This yields

$$\|e\| = O\!\left(K^{\frac{d}{2}-\alpha}\right) + O\!\left(K^{\frac{d}{2}}e^{-cm}\right), \tag{87}$$

which for suitably growing $m$ is strictly faster than the spatial-domain rate $O(K^{d/2})$ in Lemma 4. $\qquad\square$

**Remark 6** (Error Component Interpretation). *The spectral truncation term $O(K^{\frac{d}{2}-\alpha})$ reflects the accelerated decay granted by $\alpha > d/2$ in the Fourier domain. The parametrization term $O(K^{\frac{d}{2}}e^{-cm})$ demonstrates that, by increasing the adapter width $m$, one obtains exponential control over the low-frequency approximation error.*

**Proposition 3.3** (Quantitative Low–/High–Frequency Energy Split for PDE Solution). *Let $s > \frac{d}{2}$ and suppose $f \in C\big([0,T]; H^s(\mathbb{T}^d)\big)$, $\sup_{t\in[0,T]} \|f(t)\|_{H^s} \leq M$. Let $f(t,x) = \sum_{k\in\mathbb{Z}^d} \widehat{f}(t,k)\,e^{i\,k\cdot x}$, then for each $k \neq 0$,*

$$|\widehat{f}(t,k)| \leq M\,(1+\|k\|^2)^{-s/2}, \tag{4}$$

*and for every integer $K \geq 1$ there exists $C = C(d,s)$ such that*

$$\sum_{\|k\|>K} |\widehat{f}(t,k)|^2 \leq C\,M^2\,K^{d-2s}, \qquad \sum_{\|k\|\leq K} |\widehat{f}(t,k)|^2 = \|f(t)\|_{L^2}^2 - O\big(K^{d-2s}\big). \tag{5}$$

*Proof.* By assumption one has

$$\|f(t)\|_{H^s}^2 = \sum_{k\in\mathbb{Z}^d} (1+\|k\|^2)^s \left|\widehat{f}(t,k)\right|^2 \leq M^2. \tag{88}$$

Hence for each nonzero $k$,

$$\left|\widehat{f}(t,k)\right|^2 \leq M^2\,(1+\|k\|^2)^{-s}, \quad \left|\widehat{f}(t,k)\right| \leq M\,(1+\|k\|^2)^{-s/2}. \tag{89}$$

Define the high–frequency tail

$$A_K(t) = \sum_{\|k\|>K} \left|\widehat{f}(t,k)\right|^2 \leq M^2 \sum_{\|k\|>K} (1+\|k\|^2)^{-s}. \tag{90}$$

Partitioning $\{k : \|k\| > K\}$ into shells $m-1 < \|k\| \leq m$ and writing $\mathcal{N}(r) = \#\{k : \|k\| \leq r\}$, we get

$$\sum_{\|k\|>K} (1+\|k\|^2)^{-s} \leq \sum_{m=\lfloor K\rfloor+1}^{\infty} \big[\mathcal{N}(m) - \mathcal{N}(m-1)\big]\,(1+(m-1)^2)^{-s}. \tag{91}$$

Since $\mathcal{N}(m) - \mathcal{N}(m-1) \leq C_d\,m^{d-1}$ and $(1+(m-1)^2)^{-s} \leq (m-1)^{-2s}$, one obtains

$$\sum_{\|k\|>K} (1+\|k\|^2)^{-s} \leq C_d \sum_{m=\lfloor K\rfloor+1}^{\infty} m^{d-1-2s} \leq C_d \int_K^{\infty} r^{d-1-2s}\,dr = \frac{C_d}{2s-d}\,K^{d-2s}. \tag{92}$$

It follows that

$$A_K(t) \leq \frac{C_d}{2s-d}\,M^2\,K^{d-2s}. \tag{93}$$

Finally, Parseval's identity gives

$$\sum_{\|k\|\leq K} \left|\widehat{f}(t,k)\right|^2 = \|f(t)\|_{L^2}^2 - A_K(t) = \|f(t)\|_{L^2}^2 - O\big(K^{d-2s}\big), \tag{94}$$

which completes the proof. $\qquad\square$

## C  Experimental Settings and Supplementary Results

### C.1  DPOT Backbone

The *Auto-Regressive Denoising Operator Transformer* (DPOT) is a Fourier–transformer backbone designed for large-scale pre-training on heterogeneous PDE trajectories [14]. Its architecture (Figure 2 of the original paper) is factorised into four principal modules—*Patch/Positioning Embedding*, *Temporal Aggregation*, *Fourier Attention*, and *Output Projection*—which together convert raw spatiotemporal grids into operator-valued predictions while retaining full frequency information.

**Patch/Positioning Embedding Layers.**  Each input trajectory $\mathbf{u}_{<T} \in \mathbb{R}^{H \times W \times T \times C}$ is first patchified: a $P \times P$ convolution groups neighbouring cells and lifts them to a $D$-dimensional token space. Learnable positional encodings $W_p(x_i, y_j, t)$ are then added channel-wise, producing per-time-step embeddings $Z_p^t \in \mathbb{R}^{\frac{H}{P} \times \frac{W}{P} \times D}$ that remain resolution-agnostic across datasets.

**Temporal Aggregation Layers.**  To condense the temporal context, DPOT employs a **weighted temporal MLP** with complex Fourier features. For each spatial location $(i, j)$ and channel $c$, the layer forms a weighted sum

$$z_{\text{agg}}^{(i,j,c)} = \sum_{t=1}^{T} W_t^{(c)} z_p^{t,(i,j,c)} e^{-\mathrm{i}\gamma_c t}, \tag{95}$$

where $W_t^{(c)}$ and $\gamma_c$ are learnable and shared across datasets. This operation implicitly encodes time-frequency signatures that help the model infer PDE type and latent parameters from short sequences.

**Fourier Attention Layers.**  The core stack consists of $L$ **Fourier–attention layers**. Each layer lifts its input to frequency space via an FFT, applies a two-layer multi-head MLP $K_l$ to the complex coefficients, and reverts to the spatial domain with an inverse FFT before a point-wise MLP $M_l$:

$$\mathbf{z}^{(l+1)} = \mathbf{z}^{(l)} + M_l\big(\mathcal{F}^{-1}\big[K_l\big(\mathcal{F}[\mathbf{z}^{(l)}]\big)\big]\big). \tag{96}$$

This frequency-space mixing acts as an efficient global kernel integral transform and scales linearly with sequence length in practice:contentReference[oaicite:4]index=4.

**Output Projection Layers.**  Finally, a point-wise projection $Q : \mathbb{R}^D \to \mathbb{R}^{C_{\text{out}}}$ maps the latent field back to the physical variable space, optionally preceded by up-sampling or padding to match the desired resolution. Because $Q$ operates channel-wise, it is independent of grid size and can be re-used for variable-sized domains:contentReference[oaicite:5]index=5:contentReference[oaicite:6]index=6.

**Discussion.**  The modular design makes DPOT both *flexible*—handling irregular resolutions, channel counts, and temporal lengths—and *scalable*: model width $D$ and depth $L$ can be increased to the 1 B-parameter regime with near-linear FLOP growth. Moreover, the FFT–IFFT symmetry of the Fourier attention stack enables lightweight fine-tuning strategies such as our Frequency-Adaptive Adapters (Section 4), which can be inserted without modifying pre-trained weights or training schedules.

### C.2  Poseidon Backbone

The *scalable Operator Transformer* (scOT) is the backbone of Poseidon [16], designed to approximate solution operators $S(t, a)$ of time-dependent PDEs by jointly encoding the lead time $t$ and the input function $a$ in a hierarchical, multiscale vision-transformer architecture with shifted-window (SwinV2) attention. In contrast to next-step predictors, scOT directly learns the operator that maps initial data to the entire solution trajectory and supports *continuous-in-time* evaluation through time-conditioned layer normalization.

**Patch and Embedding Layers.**  Inputs $a \in C(D; \mathbb{R}^n)$ are first partitioned into non-overlapping $p \times p$ patches and linearly embedded into a $C$-dimensional latent field $v \in C(D; \mathbb{R}^C)$. This discretizes a patching operator that averages within patches and lifts to token space; the embedding is immediately normalized by a (lead-time) conditioned layer norm (see below). The construction is resolution-agnostic and serves as the interface between function-space data and transformer tokens.

**Lead-Time–Conditioned Layer Norm.** To enable real-valued time queries, scOT replaces standard layer norm LN with a *lead-time conditioned* variant that modulates the affine parameters by $t$:

$$\mathrm{LN}_{\alpha(t),\beta(t)}(v)(x) = \alpha(t) \odot \frac{v(x) - \mu_v(x)}{\sigma_v(x)} + \beta(t), \quad \alpha(t) = \alpha_t + \alpha, \ \beta(t) = \beta_t + \beta, \qquad (97)$$

with $\mu_v, \sigma_v$ the channel-wise mean and standard deviation. This simple conditioning yields continuous-in-time evaluations for $S(t, \cdot)$ within a single network.

**Shifted-Window (SwinV2) Attention Blocks.** At each scale, tokens pass through SwinV2 blocks that apply windowed multi-head self-attention within fixed spatial windows; windows are *shifted across layers* to allow global information exchange with linear complexity in the number of tokens. Each block follows a residual "attention–MLP" stack with time-conditioned layer norms on both sublayers.

**Hierarchical Encoder–Decoder with Skip Connections.** SwinV2 stages are arranged in a U-Net–style hierarchy with *patch merging* for down-scaling and *patch expansion* for up-scaling. Encoder and decoder stages at matching resolutions are connected by lightweight ConvNeXt blocks that preserve multi-scale features while keeping the bottleneck convolution-free.

**Output Recovery and Mixup.** After decoding, a recovery head reassembles the latent tokens back to the physical domain, optionally with mixup in the output space; this step is independent of the grid size and thus compatible with variable-resolution domains.

**Training Objective and all2all Supervision.** Given trajectories $\{S(t_k, a_i)\}_{k=0}^{K}$, scOT can be trained with a standard operator loss

$$L(\theta) = \frac{1}{M(K+1)} \sum_{i=1}^{M} \sum_{k=0}^{K} \| S_\theta^*(t_k, a_i) - S(t_k, a_i) \|_{L^p(D)} \quad (p = 1). \qquad (98)$$

and, crucially, with an *all2all* variant that exploits the semigroup property $S(t^*, a) = S(t^* - t, S(t, a))$ to form supervision from all intra-trajectory time pairs $(t_k, t_{\bar{k}})$ with $k \leq \bar{k}$, yielding $O(K^2)$ training pairs per trajectory and markedly improved data efficiency. At inference, scOT supports direct $t$-queries or variable-step rollout via successive applications of $S_\theta^*$.

**Discussion.** The modular design—patch embeddings, time-conditioned normalization, shifted-window attention, and multi-scale encoder–decoder—makes scOT both *flexible* (heterogeneous PDE inputs, resolutions, boundary conditions via masking) and *scalable* (depth/width and token counts). In POSEIDON, this backbone underpins large gains in sample efficiency and accuracy across diverse downstream PDE operators after pretraining on a compact set of fluid-dynamics operators.

### C.3 Fine-tuning Protocol

We begin with a DPOT backbone that was pre-trained on diverse two-dimensional PDE trajectories and adapt it to new datasets of arbitrary dimensionality.

- **Dimensional adaptation.** Only the Fourier–Attention layers and Patch-Embedding kernels are replaced with their one-, two-, or three-dimensional counterparts that match the target grid. Positional embeddings are resized with trilinear interpolation, while all remaining weights are loaded unchanged.
- **Parameter-efficient updates.** All newly added PEFT modules are initialized with (near)-zero up-projection weights, so the network initially behaves like the frozen backbone and gradually routes learning into the adapters as training proceeds.

Apart from these structural switches, the fine-tuning pipeline—optimizer, scheduler, and so forth—follows the same recipe used during pre-training, but updates only the lightweight adapter weights and a few normalisation parameters.

## C.4 Placement of PEFT Modules

PEFT Modules are inserted at four important positions of the network:

1. Patch/Positioning Embedding Layers—after each convolution that maps raw input patches into the latent space.

2. Temporal Aggregation Layers—directly after the first patchifying layer.

3. Fourier Attention Layers—before, between, and after the linear transforms operating in Fourier space.

4. Output Projection Layers—parallel to the final transposed-convolution path that reconstructs the physical field.

This arrangement grants every major transformation pathway a low-rank, trainable side route, enabling the model to specialise to new PDE systems with minimal additional parameters.

## C.5 Evaluation Metric

The *L2 Relative Error* (**L2RE**) is adopted as the sole evaluation metric. Given the test set $\mathcal{D} = \{(\mathbf{y}_i, \hat{\mathbf{y}}_i)\}_{i=1}^{N}$, L2RE is defined as

$$\text{L2RE} = \frac{1}{N} \sum_{i=1}^{N} \frac{\|\hat{\mathbf{y}}_i - \mathbf{y}_i\|_2}{\|\mathbf{y}_i\|_2}. \tag{99}$$

This ratio normalizes the prediction error by the energy of the ground-truth signal, yielding a dimension-free quantity whose smaller value indicates better performance.

## C.6 PDEBENCH 3D Compressible Navier–Stokes (CFD-3D) Dataset

The CFD-3D benchmark released with PDEBENCH [47] targets the forward prediction of turbulent, compressible flows in *three spatial dimensions*. It now comprises **three** distinct subsets, each recorded on a $128^3$ **Cartesian grid** and sharing identical solver parameters and output format:

| Subset name | Initial condition | $(\eta, \varsigma, M)$ |
| --- | --- | --- |
| NS-3D-turb | divergence–free turbulence | $(10^{-8}, 10^{-8}, 1.0)$ |
| NS-3D-rand | random Gaussian field | $(10^{-8}, 10^{-8}, 1.0)$ |
| NS-3D-rand | low–Mach random field | $(10^{-8}, 10^{-8}, 0.1)$ |

Each subset contains **100** simulation trajectories of the full compressible Navier–Stokes (CNS) equations (100)–(102).

$$\partial_t \rho + \nabla \cdot (\rho \mathbf{v}) = 0, \tag{100}$$

$$\rho(\partial_t \mathbf{v} + \mathbf{v} \cdot \nabla \mathbf{v}) = -\nabla p + \eta \nabla^2 \mathbf{v} + \left(\varsigma + \frac{\eta}{3}\right) \nabla(\nabla \cdot \mathbf{v}), \tag{101}$$

$$\partial_t \left(\varepsilon + \frac{1}{2}\rho \|\mathbf{v}\|^2\right) + \nabla \cdot \left[\left(p + \varepsilon + \frac{1}{2}\rho \|\mathbf{v}\|^2\right) \mathbf{v} - \mathbf{v} \cdot \boldsymbol{\sigma}'\right] = 0. \tag{102}$$

Every trajectory provides **21 equally-spaced snapshots** ($t \in [0, 1]$) stored as six-channel tensors $[\rho, u, v, w, p, \varepsilon] \in \mathbb{R}^{128 \times 128 \times 128 \times 6}$.

**Initial and boundary conditions.** The three subsets differ only by their *initial* velocity field and Mach number. NS-3D-turb seeds a divergence-free Kolmogorov-type spectrum, whereas NS-3D-rand and NS-3D-rand draw velocity, density and pressure perturbations from isotropic Gaussian random fields (extended from Equation (8) in 47) before adding a uniform background. Periodic boundaries are enforced in all directions, mimicking homogeneous isotropic turbulence and simplifying spectral learning methods. The random-field subsets include a fixed-Mach configuration at $M = 1.0$ and a nearly inviscid, low-Mach compressible configuration at $M = 0.1$; full specifications of these variants are provided in the official dataset card.

**Scientific and ML challenges.** Beyond the previously noted high dimensionality and shock-capturing difficulties, the extended CFD-3D benchmark now stresses surrogate models along two additional axes: (i) *initial-condition diversity* (turbulent vs. random fields) and (ii) *Mach-number variation* spanning an order of magnitude ($M = 0.1$ to $1.0$). Successful models must therefore exhibit *robust generalisation across both flow regimes and acoustic compressibility scales*.

**Splits.** Following the original protocol, we reserve 90 trajectories for training/validation and 10 for held-out testing *within each subset*. We utilize stratified sampling to preserve the subset ratios.

| Quantity | Symbol | Value | Notes |
|---|---|---|---|
| Spatial resolution | $N_x \times N_y \times N_z$ | $128^3$ | Cartesian grid |
| Time steps per run | $N_t$ | 21 | $\Delta t = 0.05$ |
| Number of runs | $N_{\text{samples}}$ | 100 | 90/10 train/test split |
| Viscosity pairs | $(\eta, \varsigma)$ | $10^{-8}, 10^{-2}$ | Two regimes |
| Mach number | $M$ | 1.0 | Isothermal EOS |
| Boundary condition | – | Periodic | All faces |
| Stored channels | – | 6 | $\rho, u, v, w, p, \varepsilon$ |

Table 7: Core statistics of the CFD-3D dataset.

## C.7 PDEBench 2D Shallow Water Equations (SWE 2D) Dataset

The SWE 2D benchmark in PDEBENCH targets forward prediction of free-surface flows in *two spatial dimensions*. Each trajectory is simulated on a $128 \times 128$ **Cartesian grid** with nonperiodic Neumann boundaries and is provided as an HDF5 array following the convention $(N, T, X, Y, V)$. The benchmark offers **1000** distinct runs and, for each run, **100** stored time steps that capture nonlinear wave fronts and shock-like features typical of shallow-water dynamics. Baseline solvers and dataset packaging follow the official PDEBENCH specification.

**Governing equations.** The two-dimensional shallow water system is supplied in conservative form with bed-slope source terms:

$$\partial_t h + \partial_x(hu) + \partial_y(hv) = 0, \tag{103}$$

$$\partial_t(hu) + \partial_x\left(hu^2 + \tfrac{1}{2}gh^2\right) + \partial_y(huv) = -g\,h\,\partial_x b, \tag{104}$$

$$\partial_t(hv) + \partial_y\left(hv^2 + \tfrac{1}{2}gh^2\right) + \partial_x(huv) = -g\,h\,\partial_y b. \tag{105}$$

where $h$ is water depth, $(u, v)$ are horizontal velocities, $b(x, y)$ is the bathymetry, and $g$ is gravitational acceleration. In this representation the prognostic variables are $(h, hu, hv)$, which makes conservation properties explicit and well defined even in the presence of discontinuities.

**Problem setup and data generation.** The dataset instantiates a radial dam-break scenario on a square domain $\Omega = [-2.5, 2.5]^2$ with an initial water mound centered at the origin,

$$h(0, x, y) = \begin{cases} 2.0, & \sqrt{x^2 + y^2} < r, \\ 1.0, & \text{otherwise,} \end{cases} \qquad u(0, x, y) = v(0, x, y) = 0,$$

where the radius $r$ is sampled uniformly per run from $[0.3, 0.7]$ to diversify initial conditions. Simulations use a finite-volume solver from `PyClaw` to generate reference trajectories that are then downsampled to the released resolution and schedule.

**Scientific and ML challenges.** SWE 2D stresses emulators through sharp fronts, wetting and drying interfaces, and reflection at nonperiodic boundaries. Accurate surrogates must conserve mass and handle momentum coupling while remaining stable over multi-step rollouts. The benchmark exposes these difficulties in a controlled setting with standardized storage and splits.

| Quantity | Symbol | Value | Notes |
|---|---|---|---|
| Spatial resolution | $N_x \times N_y$ | $128^2$ | Cartesian grid |
| Time steps per run | $N_t$ | 100 | Stored steps per trajectory |
| Number of runs | $N_{\text{samples}}$ | 1000 | 90/10 train/test split |
| Domain | $\Omega$ | $[-2.5, 2.5]^2$ | Square box |
| Boundary condition | – | Neumann | Nonperiodic |
| Initial condition | – | radial mound | $r \sim \mathcal{U}(0.3, 0.7)$; $u = v = 0$ |
| Stored channels | – | $h, , hu, , hv$ | Conservative variables |
| Solver | – | `PyClaw` | Finite-volume reference |

Table 8: Core statistics of the SWE 2D dataset in PDEBENCH.

**Splits.** Following the official protocol, we reserve 90% of runs for training and validation and 10% for held-out testing. The same split policy is used for baseline models reported in the dataset paper.

DPOT uses this SWE 2D benchmark when reporting pre-training and transfer results, which motivates our choice to adopt the same data conventions and splits.

## C.8 Bifrost Chromosphere–Corona MHD-3D Dataset

We use the publicly released *Bifrost* enhanced-network simulation distributed by the Hinode Science Data Centre Europe (ID: `en024048_hion`). It provides time-indexed 3D magnetic-field cubes $(B_x, B_y, B_z)$ on a Cartesian grid of $504 \times 504 \times 496$ that spans a physical volume of $24\,\text{Mm} \times 24\,\text{Mm} \times -2.4\,\text{Mm}$ to (numerical range) $14.4\,\text{Mm}$. The standard release contains 157 snapshots at a cadence of $10\,\text{s}$, from $t = 3850\,\text{s}$ to $t = 5410\,\text{s}$. These specifications are consistent with the original description of the run and subsequent studies that use the same source.

**Governing equations.** Bifrost advances the full resistive MHD system on a staggered mesh with high-order finite differences and explicit time stepping. The code solves, in conservative form, mass continuity, momentum balance with Lorentz force, magnetic induction, and total-energy evolution:

$$\frac{\partial \rho}{\partial t} + \nabla \cdot (\rho \mathbf{u}) = 0, \tag{106}$$

$$\frac{\partial (\rho \mathbf{u})}{\partial t} + \nabla \cdot (\rho \, \mathbf{u} \otimes \mathbf{u} - \boldsymbol{\tau}) = -\nabla p + \mathbf{J} \times \mathbf{B} + \rho \, \mathbf{g}, \tag{107}$$

$$\frac{\partial \mathbf{B}}{\partial t} = \nabla \times (\mathbf{u} \times \mathbf{B}) - \nabla \times (\eta \, \mathbf{J}), \tag{108}$$

$$\frac{\partial e}{\partial t} + \nabla \cdot (e \, \mathbf{u}) = -p \, \nabla \cdot \mathbf{u} + Q. \tag{109}$$

with $\mu_0 \mathbf{J} = \nabla \times \mathbf{B}$. Boundary treatment uses ghost zones with problem-dependent conditions; radiation, conduction, and non-equilibrium ionization are included through $Q$ and the equation of state.

**Initial and boundary conditions.** The simulation represents an enhanced network with two opposite magnetic polarities separated by about $8\,\text{Mm}$ at the photosphere. Convective driving shears and braids the field, producing realistic chromosphere–corona coupling. Lateral and top boundaries are nonperiodic in the production run used by `en024048_hion`, implemented through ghost zones in Bifrost.

**Scientific and ML challenges.** MHD-3D stresses operator learners through strong anisotropy along field lines, steep gradients near the photosphere, and nonperiodic boundaries that complicate spectral assumptions. Accurate surrogates must reconstruct 3D structure from minimal boundary information and remain stable across height with respect to physically derived metrics such as $|\mathbf{B}|$ and $|\mathbf{J}|$.

**Splits and extreme-scarcity protocol.** To probe learning under severe data scarcity, we treat each time index as a trajectory and select **24** snapshots for training. The remaining snapshots are held out

Table 9: Core statistics of the MHD-3D dataset used in our study.

| Quantity | Symbol | Value | Notes |
|---|---|---|---|
| Spatial resolution | $N_x \times N_y \times N_z$ | $504 \times 504 \times 496$ | Cartesian grid |
| Physical domain | $\Omega$ | $24 \times 24 \times 16.8 \,\mathrm{Mm}^3$ | $z \in [-2.4\,\mathrm{Mm},\ 14.4\,\mathrm{Mm}]$ |
| Snapshots per run | $N_t$ | 157 | 10 s cadence; $t \in [3850\,\mathrm{s},\ 5410\,\mathrm{s}]$ |
| Stored channels | — | $B_x, B_y, B_z$ | Magnetic field |
| Boundary condition | — | Nonperiodic | Ghost-zone implementation |
| Downsampled target | — | $504 \times 504 \times 99$ | Height subsampling |
| Train set (ours) | $N_{\text{train}}$ | 24 | Extreme scarcity |
| Source archive | — | `en024048_hion` | Hinode SDC Europe |

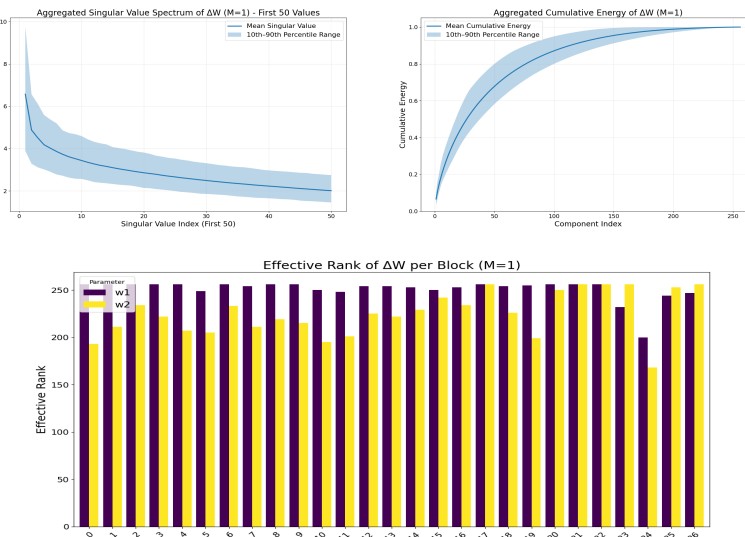

Figure 6: **Spectral diagnostics after full-rank fine-tuning at** $M = 1.0$. Top row: aggregated singular-value spectrum (left) and cumulative-energy curve (right). Bottom: block-wise effective ranks of $\Delta W$.

for testing. Unless otherwise noted, inputs are the bottom boundary magnetogram ($504 \times 504$) and targets are the downsampled ($504 \times 504 \times 99$) interior volume for the same time index, following the GL-FNO data interface.

**Provenance and access.** The `en024048_hion` cubes are curated at Hinode SDC Europe; the underlying simulation is documented by Carlsson et al. [4]. Recent ML work on coronal-field reconstruction from this source provides a consistent pre-processing recipe and confirms the 3D geometry and cadence figures listed above.

## C.9 Spectral Diagnostics of $\Delta W$ after Full-Rank Fine-Tuning

To assess the intrinsic rank of the updates obtained via unconstrained fine-tuning, we perform *full-rank* adaptation of the 1 B-parameter DPOT-H backbone on the 3D-NS random-initial-condition datasets at Mach numbers $M = 1.0$ and $M = 0.1$. After convergence, we extract the complex-valued Fourier Attention Layer weights (real and imaginary concatenated) $\{\mathbf{W}_{k,p}^{\text{ft}}\}$ for each block $k \in [0, 26]$ and projection $p \in \{w_1, w_2\}$, compute the deltas

$$\Delta \mathbf{W}_{k,p} = \mathbf{W}_{k,p}^{\text{ft}} - \mathbf{W}_{k,p}^{\text{pre}}, \tag{110}$$

flatten each $\Delta \mathbf{W}_{k,p}$ to a matrix, and perform singular-value decomposition. We define the *effective rank* as the number of singular values $\sigma_i$ satisfying $\sigma_i \geq 0.01\,\sigma_1$.

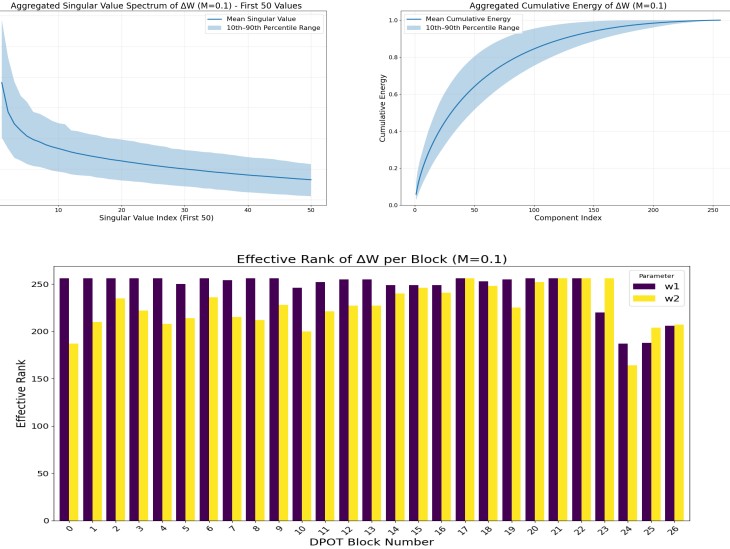

Figure 7: **Spectral diagnostics after full-rank fine-tuning at** $M = 0.1$. Top row: aggregated singular-value spectrum (left) and cumulative-energy curve (right). Bottom: block-wise effective ranks of $\Delta W$.

**Diagnostics at** $M = 1.0$. Upper left panel in Figure 6 reveals a shallow spectral decay, while upper right panel shows that only $\sim 80\%$ of the Frobenius norm is captured by the first 100 modes. Bottom Panel further indicates that nearly every block requires $\geq 240$ modes (of 256), confirming the *high-rank* nature of the update and the large approximation gap faced by rank-constrained adaptation such as LoRA (see Proposition 3.1).

**Diagnostics at** $M = 0.1$. Although the low-Mach setting leads to a slightly steeper spectral decay (upper left panel in Figure 7) and a faster accumulation of energy (upper right panel), the transformation remains far from low-rank: capturing $90\%$ of the energy still requires $\sim 140$ modes for $w_1$ and $\sim 103$ modes for $w_2$. Bottom Panel shows that the down-projection pathway retains an average effective rank of $\approx 246$, while the up-projection weights still average $\approx 226$ modes, underscoring that the intrinsic update is *still* high-rank. Consequently, even at $M = 0.1$ rank-constrained linear adapters suffer from an irreducible spectral bias.

Table 10 summarizes the effective-rank statistics of $\Delta W$ at $M = 1.0$ and $M = 0.1$, confirming that both Mach regimes yield inherently high-rank updates. Likewise, Table 11 reports the number of singular components required to capture $90\%$ of the total energy: at $M = 1.0$, $\approx 126.6$ (resp. $97.8$) modes are needed for $w_1$ (resp. $w_2$), compared to $\approx 138.6$ and $\approx 103.3$ at $M = 0.1$. Although compressible flows at $M = 0.1$ exhibit slightly steeper spectral decay, the updates remain far from low-rank. These diagnostics substantiate that full-rank fine-tuning induces intrinsically high-rank transformations, thereby imposing an irreducible spectral bias on rank-constrained linear adapters.

| Parameter | $M = 1.0$ | | | | $M = 0.1$ | | | |
| | Mean | Std. | Min | Max | Mean | Std. | Min | Max |
|---|---|---|---|---|---|---|---|---|
| $w_1$ | 250.7 | 11.4 | 200 | 256 | 245.9 | 20.3 | 187 | 256 |
| $w_2$ | 225.0 | 23.6 | 168 | 256 | 225.7 | 23.0 | 164 | 256 |

Table 10: Effective-rank statistics of $\Delta W$ whose dimension is 256 at $M = 1.0$ and $M = 0.1$ (across all Fourier-Attention blocks).

### C.10 RMSE Comparison for MLP Adapter vs. Low-Rank Truncation

The numerical results reported in the main text were obtained on a diagnostic set built from the *first Fourier-Attention block* of DPOT. Specifically, we collect $N = 200\,000$ real Fourier activations

| Parameter | $M = 1.0$ | | | | $M = 0.1$ | | | |
|---|---|---|---|---|---|---|---|---|
| | Mean | Std. | Min | Max | Mean | Std. | Min | Max |
| $w_1$ | 126.6 | 20.4 | 67 | 155 | 138.6 | 27.6 | 65 | 170 |
| $w_2$ | 97.8 | 20.8 | 52 | 138 | 103.3 | 26.5 | 52 | 154 |

Table 11: Number of singular components required to reach $90\%$ cumulative energy at $M = 1.0$ and $M = 0.1$.

$H \in \mathbb{R}^{N \times d}$ and compute their targets $Y = H\,\Delta W^\top$, where $\Delta W$ is the exact full-rank weight update after fine-tuning. The data are split $90/10\%$ into training and validation subsets before fitting either surrogate. Table 12 lists the root-mean-square error (RMSE, reported in $10^{-2}$ units) for all budgets considered. Consistent with the curves in Figure 2, the two-layer MLP adapter dominates the low-rank SVD baseline across the entire budget spectrum and for both Mach numbers.

| **Mach** | **Method** | 4 | 8 | 16 | 32 | 64 | 128 |
|---|---|---|---|---|---|---|---|
| $M = 1.0$ | Adapter (MLP) | 30.48 | 22.05 | 17.64 | 6.02 | 5.70 | **5.09** |
| | Low-Rank Trunc. | 29.10 | 27.99 | 26.08 | 22.06 | 16.83 | 8.96 |
| $M = 0.1$ | Adapter (MLP) | 24.76 | 27.67 | 18.38 | 8.43 | **4.04** | 4.46 |
| | Low-Rank Trunc. | 22.95 | 21.92 | 20.23 | 17.37 | 13.53 | 7.34 |

Table 12: Held-out RMSE ($\times 10^{-2}$) for each parameter budget. Adapter budgets correspond to hidden widths $m$; low-rank budgets correspond to SVD ranks $r$.

**Implications.** Because the MLP is trained *directly* on the $(H, Y)$ mapping it can exploit non-linear interactions in the representation space that any linear low-rank approximation of $\Delta W$ must ignore. The result corroborates our spectral analysis: even aggressive rank truncation leaves a non-negligible error floor, whereas a modest non-linear adapter is able to emulate the full-rank update with far fewer tunable parameters. This finding further reinforces the case for Adapter-style PEFT in operator learning tasks characterized by pronounced physical complexity.

### C.11 FLOPs and Inference Time for Experiments in Section 5

| Scheme | FLOPs (G) | Step Inference Time (ms) |
|---|---|---|
| AdaLoRA [58] | 543.384 | 75.631 |
| HydraLoRA [48] | 547.039 | 155.784 |
| Prompt Tuning [20] | 540.838 | 28.649 |
| Vanilla Adapter [17] | 547.469 | 81.823 |
| FiLM Adapter [43] | 548.318 | 93.676 |
| RandLoRA [1] | 545.458 | 73.432 |
| LoRA [18] | 551.008 | 71.651 |
| F-Adapter **(Ours)** | 548.531 | 90.383 |
| SVFT [25] | 630.880 | 93.026 |
| Chebyshev Adapter | 554.797 | 268.022 |
| Fourier Adapter | 546.849 | 1449.544 |
| WaveAct Adapter | 547.469 | 92.694 |
| Full Fine-Tuning | 540.838 | 27.427 |

Table 13: Computational cost and single-step latency of parameter-efficient fine-tuning strategies.

Table 13 presents a comparative overview of the computational overhead incurred by each PEFT scheme in the *3D NS Rand* $M = 1.0$ experiment. Our **F-Adapter** executes $548\,\text{G}$ FLOPs—only about 0.2% more than the Vanilla Adapter and comfortably within the typical LoRA budget—while yielding a single-step inference latency of $90\,\text{ms}$, well below the $100\,\text{ms}$ threshold commonly regarded

| Scheme | RMSLE$_E$ ↓ | RelErr$_E$ ↓ | % Param |
|---|---|---|---|
| Vanilla Adapter | 0.9186 | 10.55% | 1.16% |
| LoRA | 1.9356 | 435.09% | 1.37% |
| F–Adapter (Ours) | **0.9095** | **6.12%** | 1.91% |
| Full Fine–Tuning | 0.3208 | 0.21% | 100% |

Table 14: Spectrum–level accuracy and parameter efficiency. F–Adapter achieves the best spectral fidelity among PEFT methods while retaining a small parameter footprint.

as interactive. Although Prompt Tuning and full fine-tuning achieve slightly lower latencies (28 ms and 27 ms, respectively), the runtime premium of F-Adapter is modest and offset by its richer representational capacity. Crucially, our approach is an order of magnitude faster than hydra-style LoRA or higher-order spectral adapters specifically designed for Fourier domain, underscoring the efficiency of its frequency-adaptive design. Overall, F-Adapter strikes a favorable balance between computational cost and adaptation power, making it a practical drop-in replacement for existing adapter families in latency-sensitive scenarios.

## C.12  Spectral Analysis of PEFT Methods on DPOT

**Setup.**  We evaluate how well different PEFT methods recover multi–scale **3D turbulence** in 3D NS experiment by comparing the predicted isotropic kinetic–energy spectrum against DNS on the test set, and by inspecting 3D visualizations of velocity magnitude. For spectra, we compute $E(k)$ from the three velocity components using a 3D FFT and shell averaging in wavenumber space; prediction and DNS are processed by the same pipeline.

**Metrics.**  To quantify agreement across scales we report two spectrum–level metrics.

**(i) Root mean square logarithmic error (RMSLE) of the spectrum**

$$\text{RMSLE}_E = \sqrt{\frac{1}{N}\sum_{i=1}^{N}\Big(\log_{10} E_{\text{pred}}(k_i) - \log_{10} E_{\text{DNS}}(k_i)\Big)^2}, \qquad (111)$$

where the sum runs over wavenumber shells $\{k_i\}_{i=1}^N$. This measures shell–wise discrepancy on a logarithmic scale so that low and high $k$ bands contribute comparably.

**(ii) Relative error of the total kinetic energy**

$$E_{\text{tot}} = \int_0^{k_{\max}} E(k)\,\mathrm{d}k, \qquad \text{RelErr}_E = \frac{|E_{\text{tot,pred}} - E_{\text{tot,DNS}}|}{E_{\text{tot,DNS}}} \times 100\%. \qquad (112)$$

The integral equals the domain–averaged turbulent kinetic energy up to a constant factor, so this metric captures conservation of total energy content.

**Findings.**  Figure 8 presents $E(k)$ on logarithmic axes together with the DNS curve and a $k^{-5/3}$ reference slope for the inertial range. F–Adapter tracks the DNS spectrum closely over a broad band of $k$ and preserves the correct decay at higher wavenumbers. Vanilla Adapter exhibits a noticeable deficit in mid–to–high $k$. LoRA underestimates energy by orders of magnitude across most shells and shows noisy behavior at the largest $k$, consistent with the very large $\text{RelErr}_E$.

## C.13  Discussion on Different Types of Adapters for Fourier Domain

**Chebyshev Adapter**  Motivated by prior work [52] which leverages the frequency-domain expressivity of Chebyshev polynomials within FNO, we propose the Chebyshev KAN Adapter (Chebyshev Adapter). It utilizes the spectral expressivity of Chebyshev-based Kolmogorov–Arnold Networks[44] by replacing the standard linear up-projection with a ChebyKAN layer. Given an input

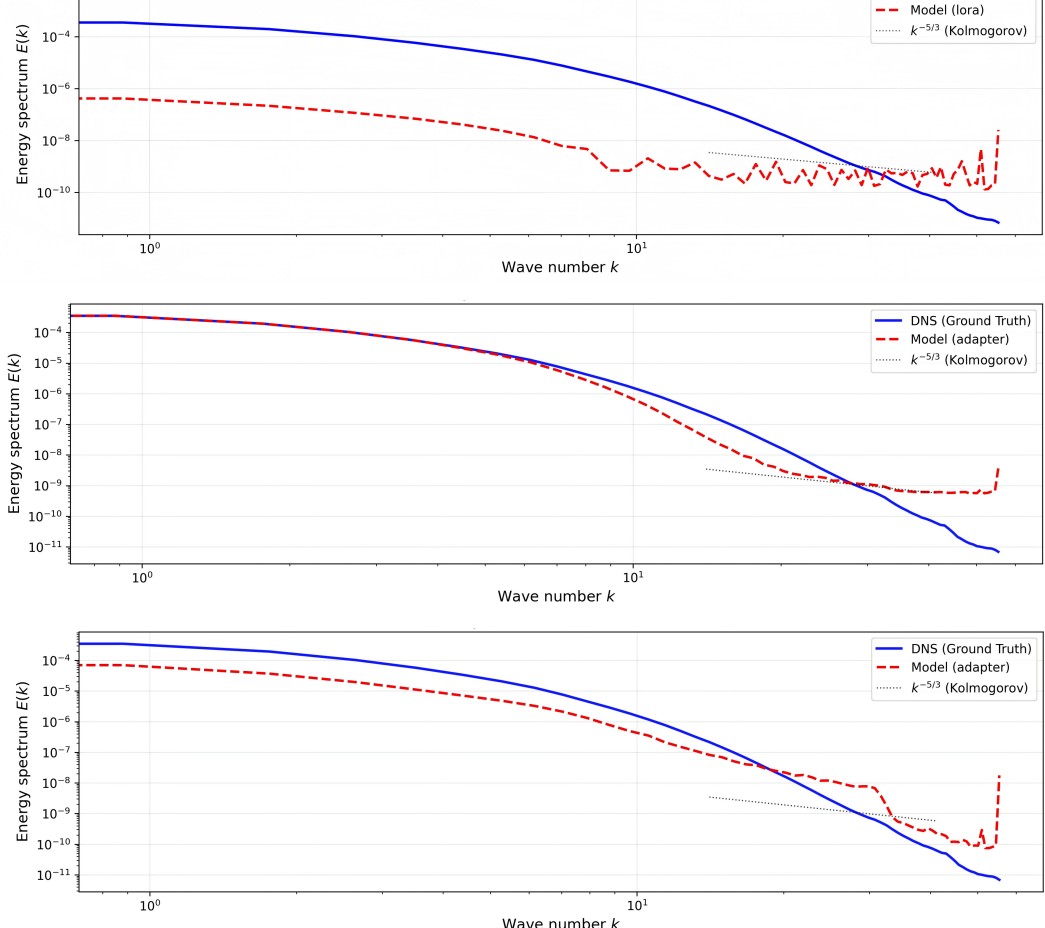

Figure 8: Energy spectra $E(k)$ on logarithmic axes for three PEFT methods, each compared to DNS (blue) and a $k^{-5/3}$ reference slope (gray). From top to bottom: LoRA, F-Adapter (Ours), and Vanilla Adapter. F-Adapter follows the DNS spectrum more closely across a broad range of wavenumbers, while Vanilla Adapter shows a deficit at mid-to-high $k$ and LoRA substantially underestimates energy with noisy behavior at large $k$.

activation $\mathbf{x} \in \mathbb{R}^{d_{\text{in}}}$, the Chebyshev Adapter computes

$$\mathbf{z} = \tanh(\mathbf{W}_{\text{down}}\mathbf{x} + \mathbf{b}_{\text{down}}), \tag{113}$$

$$y_k = \sum_{i=1}^{d_{\text{bottleneck}}} \sum_{n=0}^{N} C_{k,i,n} T_n(\tilde{z}_i), \quad k = 1, \ldots, d_{\text{in}}, \tag{114}$$

where $\mathbf{W}_{\text{down}} \in \mathbb{R}^{d_{\text{bottleneck}} \times d_{\text{in}}}$ is the learnable down-projection, $\tilde{z}_i = \tanh(z_i)$ normalizes into $[-1, 1]$, and $T_n(x) = \cos(n \arccos(x))$ is the $n$-th Chebyshev polynomial of the first kind. The coefficient tensor $C \in \mathbb{R}^{d_{\text{in}} \times d_{\text{bottleneck}} \times (N+1)}$ is learned end-to-end. Finally, a residual connection with learnable scalar $\alpha$ restores the original dimension:

$$\text{Chebyshev-Adapter}(\mathbf{x}) = \alpha \, \mathbf{y} + \mathbf{x} \,. \tag{115}$$

We initialize $\mathbf{W}_{\text{down}}$ by Kaiming uniform and set all $C_{k,i,n} = 0$ to start training from the identity mapping. The scalar $\alpha$ is also learnable, allowing the model to adaptively control the adapter's contribution.

The Chebyshev Adapter leverages the enhanced Chebyshev–KAN Layer to boost approximation power without the dense spline-grid storage required by standard Kolmogorov–Arnold Networks [30].

**Fourier Adapter**  Motivated by the amortised Fourier–kernel formulation of Xiao et al. [52] and the expressive FourierKAN layer of Xu et al. [53], we introduce the *FourierKAN Adapter* (Fourier Adapter) as a frequency-domain alternative to the vanilla bottleneck Adapter used in LOMs. Given an input activation $\mathbf{x} \in \mathbb{R}^{d_{\mathrm{in}}}$, the module first performs a linear dimension reduction

$$\mathbf{z} = \sigma\big(\mathbf{W}_{\mathrm{down}}\mathbf{x} + \mathbf{b}_{\mathrm{down}}\big), \qquad \mathbf{W}_{\mathrm{down}} \in \mathbb{R}^{d_{\mathrm{bottleneck}} \times d_{\mathrm{in}}}, \tag{116}$$

where $\sigma(\cdot)$ denotes GELU unless stated otherwise. To restore the original width we replace the standard linear up-projection with a *FourierKAN* layer that expands each scalar $z_i$ into a truncated Fourier series of order $K$:

$$y_k = \sum_{i=1}^{d_{\mathrm{bottleneck}}} \sum_{n=1}^{K} \Big( A_{k,i,n} \cos(nz_i) + B_{k,i,n} \sin(nz_i) \Big), \qquad k = 1, \dots, d_{\mathrm{in}}, \tag{117}$$

$$\mathbf{y} = \big(y_1, \dots, y_{d_{\mathrm{in}}}\big)^{\top}. \tag{118}$$

Here the learnable coefficients $A, B$ lie in $\mathbb{R}^{d_{\mathrm{in}} \times d_{\mathrm{bottleneck}} \times K}$. Because cos and sin are $2\pi$-periodic and globally supported, Eqs. (117)–(118) endow the adapter with a strong inductive bias for periodic, high-frequency phenomena that commonly arise in PDE spectra, while avoiding the dense spline grids required by classical KANs [30]. The series order $K$ is typically $\leq 256$ to curb aliasing and memory, yielding an $\mathcal{O}\big(d_{\mathrm{in}}d_{\mathrm{bottleneck}}K\big)$ cost.

A learnable LayerNorm followed by residual scaling finishes the block:

$$\mathrm{Fourier\text{-}Adapter}\big(\mathbf{x}\big) = \alpha \, \mathrm{LN}\big(\mathbf{y}\big) + \mathbf{x}, \tag{119}$$

where learnable parameter $\alpha$ is initialized to 0, so training begins from the identity map. We initialize $(A, B)$ with $\mathcal{N}\big(0, \, K^{-1/2}d_{\mathrm{bottleneck}}^{-1/2}\big)$ and attenuate high frequencies by $(n+1)^{-2}$ to ensure smooth scalar functions at start-up, following our implementation practice.

The Fourier Adapter offers parameter efficiency comparable to F-Adapter while directly modelling spectral bases; however, its global trigonometric kernels incur greater FLOPs and memory (Table 13) and can amplify aliasing when $K$ is large, echoing the empirical findings in Table 2. Nonetheless, for tasks dominated by periodic boundary conditions or sharp oscillations, it serves as a principled, physics-aware drop-in replacement for projection-based adapters.

**WaveAct Adapter**  Building on the learnable wavelet-based activation proposed in Zhao et al. [60] and the success of wavelet transforms in operator learning [50], we devise the *WaveAct-Activated Adapter* (WaveAct Adapter). Unlike functional–basis adapters that alter the projection layers themselves, WaveAct Adapter keeps the standard bottleneck architecture but replaces the pointwise non-linearity with a parameter-efficient WaveAct gate that superposes local sine and cosine responses. Formally, for an input activation $\mathbf{x} \in \mathbb{R}^{d_{\mathrm{in}}}$ we compute

$$\mathbf{z} = \mathbf{W}_{\mathrm{down}}\mathbf{x} + \mathbf{b}_{\mathrm{down}}, \qquad \mathbf{W}_{\mathrm{down}} \in \mathbb{R}^{d_{\mathrm{bottleneck}} \times d_{\mathrm{in}}}, \tag{120}$$

$$\tilde{z}_i = a \sin(z_i) + b \cos(z_i), \qquad i = 1, \dots, d_{\mathrm{bottleneck}}, \tag{121}$$

$$y_k = \sum_{i=1}^{d_{\mathrm{bottleneck}}} \big[\mathbf{W}_{\mathrm{up}}\big]_{k,i} \tilde{z}_i + b_k^{\mathrm{up}}, \qquad k = 1, \dots, d_{\mathrm{in}}, \tag{122}$$

where $a, b \in \mathbb{R}$ are *two* learnable, shared scalars that modulate the sine / cosine mixture, and $\mathbf{W}_{\mathrm{up}} \in \mathbb{R}^{d_{\mathrm{in}} \times d_{\mathrm{bottleneck}}}$. WaveAct thus provides a compact spectral gate whose frequency content is dynamically tuned during training, requiring only 2 extra parameters irrespective of width.

Finally, a residual path equipped with a learnable gain $\alpha$ restores the original dimensionality:

$$\mathrm{WaveAct-Adapter}(\mathbf{x}) = \alpha \, \mathbf{y} + \mathbf{x}, \qquad \alpha \in \mathbb{R}. \tag{123}$$

We set $(a, b) = (1, 1)$ to start from an identity-like activation ($\sin + \cos \simeq 1$ near the origin). Following Houlsby et al. [17], $\mathbf{W}_{\mathrm{up}}$ and its bias are initialised at zero so that $\alpha = 0$ yields an exact identity map at the beginning of training; $\mathbf{W}_{\mathrm{down}}$ follows Kaiming-uniform initialisation. Eq. (121) equips each bottleneck coordinate with an adaptive wavelet kernel that can synthesise both low- and high-frequency components, while preserving the memory- and FLOP-profile of the vanilla adapter

| $p$ | $r_{\min}$ | $r_{\max}$ | $B$ | L2RE | % Param | FLOPs (G) | $B_1$ dim | $B_2$ dim | $B_3$ dim | $B_4$ dim | $B_5$ dim | $B_6$ dim |
|---|---|---|---|---|---|---|---|---|---|---|---|---|
| 2 | 4 | 16 | 4 | 0.4523 | 1.91% | 548.5307 | 13 | 8 | 5 | 4 | – | – |
| 2 | 4 | 16 | 6 | 0.4509 | 2.45% | 548.7430 | 14 | 10 | 8 | 6 | 4 | 4 |
| 2 | 8 | 32 | 4 | 0.4191 | 3.40% | 555.3716 | 22 | 11 | 8 | 8 | – | – |
| 1 | 16 | 32 | 4 | 0.4203 | 4.38% | 556.8579 | 29 | 23 | 17 | 11 | – | – |
| 4 | 16 | 64 | 4 | 0.3885 | 6.76% | 569.9026 | 44 | 23 | 16 | 16 | – | – |
| 1 | 16 | 64 | 4 | 0.4152 | 8.00% | 572.8753 | 58 | 46 | 34 | 22 | – | – |

Table 15: Performance and configuration across bandwidth settings.

(Table 4). Empirically, WaveAct Adapter matches F-Adapter in memory usage and runtime yet trails slightly in L2RE (Table 5), suggesting that wavelet activations alone are insufficient to fully model the extreme high-frequency dynamics present in the Fourier domain. Nevertheless, its negligible parameter overhead and strong locality make it an attractive drop-in replacement when compute budgets are tight or periodicity is weak.

### C.14 Ablation Study over Diverse Hyperparameter Settings for F-Adapter

We conducted extensive ablation studies on diverse hyperparameter settings using DPOT and F-Adapter on the 3D-Turbulance dataset.

Results in Table 15 indicate that hyperparameters primarily influence performance by modulating adapter capacity allocation across frequency bands. This adjustment effectively governs the model's overall capacity. Crucially, hyperparameters do not directly affect performance. Their impact is mediated through capacity allocation. Consequently, selecting appropriate capacity based on computational resource constraints enables predictable performance outcomes. The magnitude of this impact remains relatively limited. This finding aligns with the insight presented in Table 1 which shows that adapters nearly obey the scaling law. Our Band-Specific Bottleneck Allocation framework in Equation (6) maintains robust generalization across diverse tasks, while hyperparameters retain flexibility to accommodate available computational resources.

### C.15 Design Details of the Frequency-Based Capacity Allocation Paradigm in the Transformer-Based Poseidon Backbone

**Motivation.** Poseidon's backbone scOT learns solution operators $S(t, a)$ with time-conditioned layer normalization and a hierarchical shifted-window attention stack. Our goal is to inject frequency awareness without rewriting the model to operate in Fourier space. We estimate frequency content per layer, allocate a capacity budget across frequency bands, and realize the budget with two parameter-efficient routes: *F-Adapter* and *F-LoRA*. The design preserves Poseidon's native spatial pipeline and continuous-in-time interface.

**Preliminaries on frequency signals.** Neural operator models such as FNO expose frequency channels through explicit spectral layers. In our transformer-based Poseidon backbone we keep all computations in the native spatial domain and recover frequency cues with lightweight probes: adjacent-token differences provide a proxy for high- versus low-frequency content in `Linear` layers, and a local real 2D FFT on `Conv2d` outputs yields a minibatch-normalized energy spectrum. The resulting per-band energies $\tilde{E}_b \in [0, 1]$ act as data-dependent gates for our adapters. Bands are defined as concentric partitions of the frequency plane, and the capacity assigned to each band follows the same rule as Equation (6), which allocates larger bottlenecks to low frequencies while reserving nonzero capacity for higher bands; the same schedule parameterizes ranks in F-LoRA.

**Frequency estimation on Poseidon.** We keep the base scOT layers intact. (i) For a **Linear** layer, we treat the token axis as a short 1D sequence, compute adjacent-token differences, and convert their magnitude to a scalar frequency score per example, which we softmax into band weights $\pi_b$. (ii) For a **Conv2d** layer inside Poseidon's downsample or upsample paths, we run a *local real 2D FFT* on the convolution output, accumulate power within each annular band, and normalize to $\pi_b$. The estimates are used only to gate adapters; the base layer remains purely spatial.

**F-Adapter.**    F-Adapter attaches a bank of lightweight per-band adapters to each target layer while freezing the base weights.

$$y_{\text{base}} = \mathcal{L}(x), \qquad y_b = \mathcal{A}_b(y_{\text{base}}), \qquad y = y_{\text{base}} + \sum_{b=0}^{B-1} \pi_b \, y_b.$$

Here $\mathcal{L}$ is the original Linear or Conv2d, and $\mathcal{A}_b$ is a bottleneck MLP for Linear layers or a $1{\times}1$ conv bottleneck for Conv2d layers with width $d_b$. The per-band outputs are combined by the data-dependent weights $\pi_b$. Only the adapter parameters are trainable. This keeps inference identical to the base scOT path plus a small residual branch and does not alter Poseidon's time conditioning.

**F-LoRA.**    F-LoRA keeps the same frequency banding and gating but replaces each bottleneck adapter with LoRA-style low-rank updates that live on the frozen weight path. For a Linear weight $W \in \mathbb{R}^{m \times n}$,

$$\mathcal{L}_{\text{F-LoRA}}(x) \;=\; Wx \;+\; \sum_{b=0}^{B-1} \pi_b \, \alpha \, A_b B_b x, \qquad A_b \in \mathbb{R}^{m \times r_b}, \; B_b \in \mathbb{R}^{r_b \times n},$$

with trainable $A_b, B_b$ and a fixed scale $\alpha$. The rank $r_b$ follows the same capacity rule as $d_b$, which concentrates rank on low frequencies while keeping nonzero rank for higher bands. For Conv2d we use equivalent $1{\times}1$ factorizations per band in channel space. F-LoRA inherits the strong optimization behavior of LoRA on transformer backbones and maintains a small trainable footprint.

**Implementation notes.**    Both mechanisms freeze the original Poseidon weights. F-Adapter attaches per-band residual branches whose last projection is initialized at zero to avoid training instability at warm start. F-LoRA initializes $B_b$ from a truncated normal and $A_b$ at zero, which recovers the base model at step zero as in standard LoRA. The energy estimator and the gating $\pi_b$ are differentiable but do not introduce global FFTs, so training throughput is close to that of the base model. The loss follows Poseidon's operator objective on sampled times with $L^1$ norm, which keeps supervision aligned with operator learning rather than single-step forecasting.

## D    Limitations

Our theoretical guarantees rest on a *low–frequency-dominance* condition—namely, that the Fourier energy spectrum of the target operator decays sufficiently fast so that most variance is captured by the first few modes. This premise is supported across a broad class of dissipative PDEs, including incompressible and compressible Navier–Stokes, reaction–diffusion, shallow–water, and advection–diffusion systems, all of which exhibit steep inertial-range energy spectra. Nevertheless, its universality remains to be fully established—strongly non-linear, multi-physics flows (e.g., MHD turbulence or reactive plasmas) may display flatter spectra that subtly stretch the separability premise in Proposition 3.2- 3.3. A theoretically rigorous characterization of how non-monotone or multi-modal spectra influence our approximation error bounds, and whether adaptive frequency-aware capacity allocation can re-establish similar guarantees in these settings, remains an open and compelling direction for future work.

## E    Broader Impacts

The proposed frequency–adaptive adapter framework lowers the computational and memory footprint required to fine-tune large operator models (LOMs) for complex partial-differential-equation systems, potentially democratizing high-resolution scientific forecasting in climate science, aerospace design, and renewable-energy optimization by enabling researchers with modest hardware to customise state-of-the-art solvers. By concentrating learnable capacity on the most energetic spectral modes, our method also reduces training energy consumption relative to full fine-tuning, contributing to greener AI practice. At the same time, accelerated surrogate models for fluid and plasma dynamics could be misused for strategic weapon design or proprietary industrial processes; we therefore commit to releasing code under a research-only licence and to incorporating provenance logging to

discourage dual-use. Finally, because adapter-based surrogates may still propagate modelling bias when extrapolating beyond their training spectra, we encourage downstream practitioners to couple our models with established uncertainty-quantification workflows and to validate predictions against trusted baselines before deployment in safety-critical settings.

