# OpenReview forum: "F-Adapter: Frequency-Adaptive Parameter-Efficient Fine-Tuning in Scientific Machine Learning"
_NeurIPS.cc/2025/Conference — NeurIPS 2025 poster_

### Official Review · Reviewer_acdm · 2025-06-25

**Clarity:** 2
**Significance:** 2
**Originality:** 3
**Rating:** 4
**Confidence:** 4

**Summary:**

This paper introduces F-Adapter, a frequency-adaptive, parameter-efficient fine-tuning method tailored for foundation models in scientific machine learning. The F-Adapter allocates more capacity to low-frequency modes and less to high-frequency ones. The authors provide both theoretical analysis and empirical validation, showing that F-Adapter outperforms LoRA and other baselines on challenging 3D Navier–Stokes tasks.

**Questions:**

- Have the authors considered directly comparing the predicted velocity spectrum to the ground-truth spectrum?

- Since the proposed F-Adapter operates in the Fourier domain, it appears tailored to FNO-based architectures. Could the authors comment on the feasibility of extending this approach to non-Fourier models or more general operator learning frameworks?

- Would it be possible to include visual comparisons (e.g., slices or isosurfaces) of the predicted and ground-truth velocity fields? This would help assess whether the model captures the spatial structure of the flow beyond what L2 error metrics reveal.

**Ethical Concerns:**

["NO or VERY MINOR ethics concerns only"]

**Final Justification:**

The authors addressed most of the points I raised in my review, so I am increasing my score to 4. However, I believe a more appropriate score would be 4.5, as it is stronger than a borderline accept.

**Limitations:**

The authors have addressed some of the limitations in the appendix.

**Paper Formatting Concerns:**

/

**Quality:**

3

**Strengths And Weaknesses:**

Strengths

- The explanation of the lower bound for block-wise LoRA in Section 3.2 is clearly presented and adds valuable theoretical insight.

- The theoretical framework presented by the authors is both rigorous and well-motivated, offering an intuitive perspective. Overall, the paper stands out for its principled integration of spectral theory into adapter design. By aligning model capacity with the energy distribution of PDE solutions, the approach seem to be both physically grounded and practically effective.

- The authors demonstrate the computational efficiency of their method, alongside strong empirical results that support its practical utility.

- The ablation studies are well-designed, providing interesting insights into both the frequency-based adapter scheduling and the choice of Fourier-domain adapter types.


Weaknesses

- In the 3D Navier–Stokes setup, the DPOT-1B model is fine-tuned on a difficult dataset, and the relative errors are high ($\geq 40\%$). The paper lacks visualizations comparing predictions and ground truth, making it unclear whether the models (using either LoRA or adapter-based tuning) simply converge to the training distribution mean with limited spectral correction. Such large errors raise concerns about the limitations of the pretrained model itself, and whether the adapter's performance can be meaningfully assessed in this setting.

- Although the method is benchmarked against several baselines, only two very challenging datasets are used for fine-tuning. Given that even full fine-tuning yields high errors, it remains unclear how the method would perform on a broader range of PDE problems.

- The proposed method relies on injecting adapters in the spectral domain after FFT, meaning it is only applicable to FNO-style architectures. However, prior work [1] has shown that FNOs scale poorly with increasing training data, casting doubt on the broader applicability of this method compared to more flexible techniques like LoRA.

- The paper does not evaluate performance under extreme data scarcity, such as fine-tuning with only 20-30 trajectories. It is unclear whether the method would remain effective under such low-data regimes, which is the most interesting regime in SciML, where we face data scarcity.

- In turbulent flow modeling, capturing fine-scale structures is essential due to the multi-scale nature of the system, the energy cascade process, and the presence of extreme events. Prioritizing only the energy-dominant modes may hinder accurate prediction of critical small-scale dynamics. I recommend that the authors compare the predicted velocity spectrum with the ground-truth spectrum to assess whether the individual modes are accurately captured.

---

[1] Herde, M., Raonic, B., Rohner, T., Käppeli, R., Molinaro, R., de Bézenac, E., \& Mishra, S. (2024). Poseidon: Efficient foundation models for pdes. Advances in Neural Information Processing Systems, 37, 72525-72624.
\end{itemize}

---

> ### Author Rebuttal · Authors · 2025-07-31
>
> Thanks for the thoughtful comments!
>
> **Reply to W1 & Q3**
>
> First, we evaluated a simpler test case for the base model: the 2D shallow water equations (SWE-2D) from PDEBench. **This choice removes any bias from base model limitations when assessing PEFT methods.** The results show that F Adapter achieves the lowest relative error (≈1.6%).
>
> |Scheme on DPOT|SWE-2D L2RE|SWE-2D %Param|
> |-|-|-|
> |AdaLoRA|0.1061|0.70%|
> |HydraLoRA|0.0956|0.88%|
> |Prompt Tuning|0.1050|0.11%|
> |Vanilla Adapter|0.0902|0.48%|
> |FiLM Adapter|0.0162|0.57%|
> |RandLoRA|0.1568|1.05%|
> |LoRA|0.1081|1.40%|
> |F-Adapter(Ours)|**0.0116**|1.24%|
> |SVFT|0.0975|0.84%|
> |Full Fine-Tuning|0.0023|100%|
>
> Second, we carried out visual comparisons of predicted and ground truth velocity fields using slice plots and isosurfaces. Unfortunately, rebuttal policy prevents us from including these figures here. In broad terms, the F‑Adapter preserves the filament‑like vortical structures and reproduces both high‑ and low‑energy regions, with only small local deviations in amplitude. LoRA, in contrast, yields coarse block‑shaped patches with muted intensities and loses most fine‑scale features, suggesting a collapse toward a spatially averaged flow. These qualitative visual observations support our quantitative error analysis and show that the adapter retains considerably more spectral detail than LoRA in this challenging regime. **We will present the full set of visual results in the camera ready version.**
>
> **Reply to W2 & W4**
>
> Firstly, our original manuscript actually evaluated on three very challenging datasets (see Appendix C.5).
>
> Besides the SWE-2D introduced in our **Reply to W1 & Q3,** we now add another experiment on the 3D magnetohydrodynamic equations (MHD-3D) under extreme data scarcity. We adopt the GL-FNO[1] data processing protocol but train on only 24 trajectories.
> |Scheme on DPOT|MHD-3D L2RE|MHD-3D %Param|
> |-|-|-|
> |AdaLoRA|1.0022|0.69%|
> |HydraLoRA|0.9440|0.85%|
> |Prompt Tuning|0.9950|1.03%|
> |Vanilla Adapter|0.7226|1.16%|
> |FiLM Adapter|0.7593|1.30%|
> |RandLoRA|0.9800|1.36%|
> |LoRA|0.9845|1.37%|
> |F-Adapter(Ours)|**0.6341**|1.91%|
> |SVFT|1.0004|2.31%|
> |Full Fine-Tuning|0.4190|100%|
>
> The results show that even with such limited data, our F Adapter continues to achieve substantially higher accuracy than the alternative PEFT schemes which fail to adapt under these conditions.
>
> [1] Du, Yutao, et al. "Global-local Fourier Neural Operator for Accelerating Coronal Magnetic Field Model." 2024
>
> **Reply to W3 & Q2**
>
> **Firstly, FFTs (FNO) are still the cornerstone of most LOMs, much as Multi‑Head Attention is for Large Language Models.** With elaborate design, it can still stably scale up and unleash powerful pre-trained capabilities. Representative FFT‑based LOMs include
> 1. PreLowD (TMLR 2024), which relies on a factorized Fourier Neural Operator whose core loop is FFT, inverse FFT, and spectral convolution.
> 2. UPS (TMLR 2024), whose first stage applies FNO spectral blocks to map the physical field into a token sequence processed by a language‑model body.
> 3. CoDA‑NO (NeurIPS 2024), where on a uniform grid the key operators K, Q, V, M, and the integral operator I are all implemented with FNO components.
> 4. OmniArch (ICML 2025), which uses a Fourier encoder to move spatial field values to the frequency domain.
> 5. DPOT (ICML 2024), our main backbone, which is the only open‑source model in this family with more than 1B parameters and outperforms purely transformer‑based baselines such as MPP in their original paper.
>
> Secondly, although an FNO backbone provides direct access to frequency features, our main focus is on assigning each frequency band its own proper bottleneck dimension rather than strictly performing convolution in the frequency domain. This insight allows our F‑Adapter to extend naturally to non‑FFT architectures. On the pure transformer‑based **Poseidon** model, we estimate frequency energy for each Linear layer from adjacent‑token differences, and for each Conv2d layer we perform a local real 2‑D FFT on the convolution output to obtain an energy spectrum that guides the adapter’s weight generation. The adapter itself still operates in the native spatial domain. Capacities are allocated to bands according to their energy, following Eq(6) of our paper, which equips the model with frequency awareness. The accompanying table reports the resulting performance gains on 2D Shallow Water Equation Dataset.
> |Scheme on Poseidon|SWE-2D L2RE|%Param|
> |-|-|-|
> |Prompt Tuning|>1.0|0.07%|
> |LoRA|0.4010|2.07%|
> |RandLoRA|0.3134|2.07%|
> |Vanilla Adapter|0.6231|2.18%|
> |AdaLoRA|0.3756|2.32%|
> |HydraLoRA|0.3474|2.57%|
> |FiLM Adapter|0.4567|3.19%|
> |SVFT|0.6742|4.22%|
> |F-Adapter|0.4311|4.17%|
> |**F-LoRA**|**0.2746**|4.78%|
> |Full Fine-Tuning|0.1534|100%|
>
> We observe that adapters deliver strong results when the base model is an FNO, yet their effectiveness declines sharply on a transformer backbone. In contrast, LoRA and its variants demonstrate robust performance on transformer backbones, reflecting established best practices in fine-tuning LLMs. But our F‑Adapter still narrows this gap by significantly improving adapter performance on transformers. **Building on this insight, we introduce F‑LoRA: it preserves the frequency‑based capacity allocation of F‑Adapter while replacing the bottleneck MLP with LoRA‑style low‑rank linear updates.** F‑LoRA achieves SOTA performance across a broad suite of PEFT methods in this setting.
>
> **Reply to W5 & Q1**
>
> Below we present additional spectral results on the 3‑D Turbulence dataset.
>  To quantify how well individual scales are reproduced, we report two metrics:
>
> 1. Root mean square logarithmic error (RMSLE) of the energy spectrum
> $$
> \text{RMSLE}\_{E}
>    = \sqrt{\frac{1}{N}\sum\_{i=1}^{N}
>       \bigl(\log_{10}E_{\text{pred}}(k_i)
>            -\log_{10}E_{\text{DNS}}(k_i)\bigr)^{2}}
> $$
> We compare the predicted and DNS energy spectra shell by shell in wave number space. This metric measures their discrepancy on a logarithmic scale and ensures that both low and high wave number bands receive equal weight.
>
> 2. Relative error of the total kinetic energy
> $$
> E_{\text{tot}}
>    = \int_{0}^{k_{\max}} E(k) \mathrm{d}k,
> \qquad
> \text{RelErr}\_{E}
>    = \frac{\lvert E_{\text{tot,pred}} - E_{\text{tot,DNS}}\rvert}
>           {E_{\text{tot,DNS}}}\times 100\\%
> $$
> Here $E_{\text{tot}}$ is the area under the spectrum. Conservation of total energy confirms that this quantity is directly proportional to the volume‑averaged turbulent kinetic energy.
>
> |Scheme|$RMSLE_E$|$RelErr_E$|%Param|
> |-|-|-|-|
> |Vanilla Adapter|0.9186|10.55%|1.16%|
> |LoRA|1.936|435.09%|1.37%|
> |F-Adapter(Ours)|**0.9095**|**6.12%**|1.91%|
> |Full Fine-Tuning|0.3208|0.21%|100%|
>
> In addition, the camera‑ready version will include a plot of $E(k)$ and the DNS spectrum on logarithmic axes. **The experimental results demonstrate that our F‑adapter significantly outperforms both LoRA and the strongest competing baseline, the Vanilla Adapter, in accurately capturing the energy spectrum.**

---

> > ### Comment · Reviewer_acdm · 2025-08-04
> >
> > Thank you for the detailed response! I hope you’ll incorporate those findings into the CRV. I am increasing my score to 4.

---

> > > ### Author Response · Authors · 2025-08-04
> > >
> > > **Thank you very much for your encouraging feedback!** We greatly appreciate your support and recognition of our work. All findings discussed during the rebuttal will be carefully consolidated and reflected in the camera-ready version to ensure they are clearly presented.

---

### Official Review · Reviewer_gNLD · 2025-06-27

**Clarity:** 3
**Significance:** 3
**Originality:** 3
**Rating:** 4
**Confidence:** 4

**Summary:**

### Summary
The paper targets parameter-efficient fine-tuning (PEFT) for large Fourier-Neural-Operator (FNO) models.
1. Low-rank LoRA severely underperforms when adapting a 1 B-parameter pretrained FNO on 3-D Navier–Stokes forecasting, whereas a simple two-layer adapter converges faster and attains lower error.
2.  The authors prove a depth-amplified approximation lower-bound for stacked low-rank updates, while two-layer MLP adapters possess universal approximation capacity in Fourier layers and achieve exponentially decaying error when focused on low-frequency modes.
3.  They propose a Frequency-Adaptive Adapter (F-Adapter). Each Fourier layer is partitioned into radial frequency bands; the adapter hidden width  $ r_b = r_{\min} + (r_{\max}-r_{\min}) \Bigl(1-\tfrac{f_b}{M}\Bigr)^{p} $  assigns larger capacity to low-frequency modes carrying most PDE energy, smaller capacity to high-frequency modes.
4.  On multiple 3-D Navier–Stokes benchmarks, F-Adapter tunes \< 2 % of weights yet surpasses LoRA and other PEFT baselines, establishing new state-of-the-art L2 relative errors.

**Questions:**

### Questions for Authors
1. How would F-Adapter perform on operator networks without explicit FFTs, or on PDEs with broadband spectra (e.g., MHD)?
2.   How robust is performance to the number of bands and the exponent \(p\)? Could these be learned automatically?
3.  What is the accuracy of full-parameter fine-tuning on the same tasks, and how close does F-Adapter come?
4. Could combining LoRA (for linear layers) with non-linear adapters reduce overhead further while mitigating LoRA’s depth error?

**Ethical Concerns:**

["NO or VERY MINOR ethics concerns only"]

**Final Justification:**

The questions and concern have been addressed. I believe the sucessful application of parameter fine-tuning in operator learning is interesting.

**Limitations:**

Yes

**Quality:**

3

**Strengths And Weaknesses:**

### Strengths
1.  First systematic PEFT study for large operator models in scientific ML.
2.  Frequency-based capacity allocation is justified by spectral energy analysis and proved beneficial.
3.   Formal lower-bound for LoRA and universal-approximation result for adapters clarify why non-linear adapters excel.
4. Comprehensive benchmarks, ablations, FLOP analyses, and convergence plots demonstrate robustness.
5.  < 2 % trainable parameters with negligible FLOP increase enables low-resource adaptation of 1 B-parameter PDE solvers.

### Weaknesses
1.  Experiments cover only Fourier-based Navier–Stokes; generality to other PDEs or non-spectral backbones is untested.
2.  Band count, \(r_{\min}, r_{\max}, p\) require tuning; sensitivity analysis is brief.
3.  No full fine-tuning reference; few comparisons to alternative PEFT schemes (e.g., BitFit).
4.  Combines known components (adapters, spectral splits); methodological leap is domain-specific rather than fundamentally new.

### Novelty
Novel within SciML: the first PEFT method that exploits Fourier energy profiles to size adapters. Theoretical framing and empirical validation are original for this domain, though the underlying ideas (adapters, frequency banding) are incremental relative to broader ML literature.

---

> ### Author Rebuttal · Authors · 2025-07-31
>
> Thanks for the thoughtful comments!
>
> **Reply to W1**
>
> Thanks for your concerns. We would like to discuss our exploration on non-spectral backbones here.
>
> **Firstly, FFTs (FNO) are still the cornerstone of most LOMs, much as Multi‑Head Attention is for Large Language Models.** Representative FFT‑based LOMs include
> 1. PreLowD (TMLR 2024), which relies on a factorized Fourier Neural Operator whose core loop is FFT, inverse FFT, and spectral convolution.
> 2. UPS (TMLR 2024), whose first stage applies FNO spectral blocks to map the physical field into a token sequence processed by a language‑model body.
> 3. CoDA‑NO (NeurIPS 2024), where on a uniform grid the key operators K, Q, V, M, and the integral operator I are all implemented with FNO components.
> 4. OmniArch (ICML 2025), which uses a Fourier encoder to move spatial field values to the frequency domain.
> 5. DPOT (ICML 2024), our main backbone, which is the only open‑source model in this family with more than 1B parameters and outperforms purely transformer‑based baselines such as MPP in their original paper.
>
> Secondly, although an FNO backbone provides direct access to frequency features, our main focus is on assigning each frequency band its own proper bottleneck dimension rather than strictly performing convolution in the frequency domain. This insight allows our F‑Adapter to extend naturally to non‑FFT architectures. On the pure transformer‑based **Poseidon** model, we estimate frequency energy for each Linear layer from adjacent‑token differences, and for each Conv2d layer we perform a local real 2‑D FFT on the convolution output to obtain an energy spectrum that guides the adapter’s weight generation. The adapter itself still operates in the native spatial domain. Capacities are allocated to bands according to their energy, following Eq(6) of our paper, which equips the model with frequency awareness. The accompanying table reports the resulting performance gains on 2D Shallow Water Equation Dataset.
> |Scheme on Poseidon|SWE-2D L2RE|%Param|
> |-|-|-|
> |Prompt Tuning|>1.0|0.07%|
> |LoRA|0.4010|2.07%|
> |RandLoRA|0.3134|2.07%|
> |Vanilla Adapter|0.6231|2.18%|
> |AdaLoRA|0.3756|2.32%|
> |HydraLoRA|0.3474|2.57%|
> |FiLM Adapter|0.4567|3.19%|
> |SVFT|0.6742|4.22%|
> |F-Adapter|0.4311|4.17%|
> |**F-LoRA**|**0.2746**|4.78%|
> |Full Fine-Tuning|0.1534|100%|
>
> We observe that adapters deliver strong results when the base model is an FNO, yet their effectiveness declines sharply on a transformer backbone. In contrast, LoRA and its variants demonstrate robust performance on transformer backbones, reflecting established best practices in fine-tuning LLMs. But our F‑Adapter still narrows this gap by significantly improving adapter performance on transformers. **Building on this insight, we introduce F‑LoRA: it preserves the frequency‑based capacity allocation of F‑Adapter while replacing the bottleneck MLP with LoRA‑style low‑rank linear updates.** F‑LoRA achieves SOTA performance across a broad suite of PEFT methods in this setting.
>
> **Reply to W2**
>
> We conducted extensive ablation studies on diverse hyperparameter settings using DPOT and F-Adapter on 3D-Turbulance dataset.
> |p|r_min|r_max|B|L2RE|%Param|FLOPs(G)|B1 dim|B2 dim|B3 dim|B4 dim|B5 dim|B6 dim|
> |-|-|-|-|-|-|-|-|-|-|-|-|-|
> |2|4|16|4|0.4523|1.91%|548.5307|13|8|5|4|/|/|
> |2|4|16|6|0.4509|2.45%|548.7430|14|10|8|6|4|4|
> |2|8|32|4|0.4191|3.40%|555.3716|22|11|8|8|/|/|
> |1|16|32|4|0.4203|4.38%|556.8579|29|23|17|11|/|/|
> |4|16|64|4|0.3885|6.76%|569.9026|44|23|16|16|/|/|
> |1|16|64|4|0.4152|8.00%|572.8753|58|46|34|22|/|/|
>
> The first row details the default hyperparameter configuration applied across all experiments in our original manuscript. Results indicate that hyperparameters primarily influence performance by modulating adapter capacity allocation across frequency bands. This adjustment effectively governs the model's overall capacity. **Crucially, hyperparameters do not directly affect performance. Their impact is mediated through capacity allocation.** Consequently, selecting appropriate capacity based on computational resource constraints enables predictable performance outcomes. The magnitude of this impact remains relatively limited. This finding aligns with the insight presented in Table 1 of our original paper, which shows that adapters nearly obey the scaling law. Our Band-Specific Bottleneck Allocation framework (Eq 6) maintains robust generalization across diverse tasks, while hyperparameters retain flexibility to accommodate available computational resources.
>
> **Reply to W3**
>
> **In Table 2, we present a comprehensive comparison between full fine‑tuning, our F Adapter, and 8 alternative PEFT schemes.** This evaluation includes methods such as LoRA, Adapter, and Prompt Tuning. Across most benchmarks, all eight approaches perform better than BitFit. Since our main manuscript focuses on frequency‑domain PEFT designs and BitFit only updates bias parameters without delivering targeted improvements or new insights, we did not include BitFit as a baseline.
>
> **Reply to W4**
>
> We appreciate this perspective on methodological novelty. While our framework integrates established concepts like adapters and spectral decomposition, its core contribution lies in **the first unified theoretical and empirical framework** comprehensively addressing domain-specific fine-tuning challenges in SciML. Critically, our frequency-aware parameter allocation principle (Eq. 6) extends beyond Fourier-based architectures. When applied to transformer based models like Poseidon through F-LoRA, it still achieves SOTA results, demonstrating broader applicability beyond spectral architectures. This evidence confirms that our high-level methodology generalizes to diverse architectures and tasks.  The details have been shown in **Reply to W1.** **These findings reveal a transferable design principle: task performance in SciML is closely linked to the efficient division of labor across features of varying frequencies, regardless of the base architecture.** While instantiated here for PDE-solving, this paradigm provides actionable insights for other tasks with strong frequency dependencies.
>
> **Reply to Q1**
>
> Firstly, please refer to **Reply to W1** for the performance of F-Adapter on operator networks without explicit FFTs. Secondly, we supplement our study with a new set of comparative tests on the 2D shallow‑water equations (SWE) and the 3D magnetohydrodynamic equations (MHD). In the SWE setting, phenomena such as hydraulic jumps and strongly non‑uniform interfaces produce shocklike discontinuities and pronounced high‑wavenumber content. The MHD represents a class of PDEs characterized by broadband spectral behavior. Moreover, we only extracted 24 trajectories for HHD training, aiming to evaluate the performance under extreme data scarcity conditions.
> |Scheme on DPOT|SWE-2D L2RE|SWE-2D %Param|MHD-3D L2RE|MHD-3D %Param|
> |-|-|-|-|-|
> |AdaLoRA|0.1061|0.70%|1.0022|0.69%|
> |HydraLoRA|0.0956|0.88%|0.9440|0.85%|
> |Prompt Tuning|0.1050|0.11%|0.9950|1.03%|
> |Vanilla Adapter|0.0902|0.48%|0.7226|1.16%|
> |FiLM Adapter|0.0162|0.57%|0.7593|1.30%|
> |RandLoRA|0.1568|1.05%|0.9800|1.36%|
> |LoRA|0.1081|1.40%|0.9845|1.37%|
> |F-Adapter(Ours)|**0.0116**|1.24%|**0.6341**|1.91%|
> |SVFT|0.0975|0.84%|1.0004|2.31%|
> |Full Fine-Tuning|0.0023|100%|0.4190|100%|
>
> Our results demonstrate that F‑Adapter still achieves SOTA. Intuitively, this stems from our frequency‑adaptive allocation of model capacity. **Rather than neglecting high‑frequency features, we partition the spectrum so that all frequencies are processed more efficiently and effectively.** F‑Adapter combines three complementary mechanisms: wide channels for low‑frequency components, lightweight residual connections for high‑frequency components, and a nonlinear universal approximator to concentrate limited parameters on the most critical features. High‑frequency information, although important, is energetically sparse and can be captured with small bottleneck modules. These modules both preserve essential details and impose an implicit regularization. Consequently, F‑Adapter excels in turbulent flows and other scenarios dominated by high‑frequency dynamics.
>
> **Reply to Q2**
>
> We would like to clarify that the hyperparameters that determine the model’s dimensionality may be difficult to make adaptive or learnable, since they fix the shape of weight tensors and cannot be updated during training. Nonetheless, we conducted extensive ablation studies on these settings in our **Reply to W2**.
>
> **Reply to Q3**
>
> **Table 2** in our manuscript includes the accuracy results of full parameter fine tuning for reference, and in several cases F-Adapter attains higher performance.
>
> **Reply to Q4**
>
> This is a valuable suggestion that aligns with our interest in refining PEFT strategies for SciML. Notably, **we have already conducted preliminary experiments on frequency-adaptive LoRA specifically tailored for Transformer-based large operator models, with details available in *Reply to W1.*** We believe these frequency-adaptive LoRA components could potentially be integrated with our current framework to leverage the strengths of both linear and non-linear adaptations. However, such integration requires careful design and extensive validation to ensure compatibility and performance gains, which would demand additional time beyond the current scope. Thus, we plan to pursue this integration as a dedicated direction in our future research.

---

> > ### Comment · Reviewer_gNLD · 2025-08-08
> >
> > I appreciate the authors' detailed explanation and addtional experiments.  I have no further questions.

---

> > > ### Author Response · Authors · 2025-08-08
> > >
> > > Thank you for your supportive feedback. We truly appreciate your thorough review and are glad to have addressed all your concerns.

---

### Official Review · Reviewer_dyWR · 2025-06-30

**Clarity:** 3
**Significance:** 2
**Originality:** 2
**Rating:** 4
**Confidence:** 4

**Summary:**

This paper studies the fine-tuning problems of large operator models in scientific machine learning (SciML). It provides an empirical analysis and theoretical insights on the spectral structure of PDE solutions. On the methods and experiments side, the authors consider DPO, which uses the Fourier Neural Operator (FNO) as a backbone, as an example for analyzing the fine-tuning performance. They introduce F-Adapter, a frequency-adaptive adapter for fine-tuning. The empirical results have shown the superiority of the proposed method in terms of generalization and spectral fidelity compared to the existing fine-tuning methods.

**Questions:**

- LoRA is not designed for improving the fine-tuning performance but to improve computational efficiency. How does the computational efficiency look in the proposed F-Adaptor method compared to LoRA?
- Since the authors focus on the fine-tuning of large models, why don’t you also consider supervised fine-tuning (SFT)? There are some SFT works in SciML. [1]

[1] Chen, Wuyang, et al. "Data-efficient operator learning via unsupervised pretraining and in-context learning." Advances in Neural Information Processing Systems 37 (2024): 6213-6245.

**Ethical Concerns:**

["NO or VERY MINOR ethics concerns only"]

**Limitations:**

The authors have addressed their limitations and under-explored aspects in the paper, such as the generalizability of the premise for F-Adapter.

**Paper Formatting Concerns:**

There is no major formatting concerns.

**Quality:**

3

**Strengths And Weaknesses:**

Strengths
- This paper investigates an important problem of post-tuning in large-scale scientific machine learning.
- This paper provides an empirical analysis and theoretical insights into the fine-tuning methods in terms of the energy distribution of PDE solutions.
- This paper is well-written and well-organized.

Weaknesses
- The scope of the studied large models in SciML is relatively limited. There are a bunch of large SciML models currently, as the authors also mentioned in Related Works. Some of them are based on transformer architectures without the Fourier transform or FNO. The empirical analysis and the proposed methods are more focused on DPOT, which uses FNO as a backbone. Therefore, I am not sure if the authors still have the same observations or conclusions after trying some other transformer-based models.
- A follow-up concern is that the definition of larger operator models is kind of unclear to me. I was thinking of a large-scale model using neural operators as backbones. However, in the related work, the authors also include MPP and Poseidon, which use transformers as backbones, as large operator models. In other words, I was confused by this paragraph, especially “Building on this, …, these large-scale operator models, built upon the FNO framework,...” Many of the cited papers clearly didn’t use FNO. [1,2,3]

[1] McCabe, Michael, et al. "Multiple physics pretraining for physical surrogate models." arXiv preprint arXiv:2310.02994 (2023).

[2] Hao, Zhongkai, et al. "Dpot: Auto-regressive denoising operator transformer for large-scale pde pre-training." arXiv preprint arXiv:2403.03542 (2024).

[3] Li, Zongyi, et al. "Fourier neural operator for parametric partial differential equations." arXiv preprint arXiv:2010.08895 (2020).

---

> ### Author Rebuttal · Authors · 2025-07-31
>
> Thanks for the thoughtful comments!
>
> **Reply to W1**
>
> **Firstly, FFTs (FNO) are still the cornerstone of most LOMs, much as Multi‑Head Attention is for Large Language Models.** Representative FFT‑based LOMs include
> 1. PreLowD (TMLR 2024), which relies on a factorized Fourier Neural Operator whose core loop is FFT, inverse FFT, and spectral convolution.
> 2. UPS (TMLR 2024), whose first stage applies FNO spectral blocks to map the physical field into a token sequence processed by a language‑model body.
> 3. CoDA‑NO (NeurIPS 2024), where on a uniform grid the key operators K, Q, V, M, and the integral operator I are all implemented with FNO components.
> 4. OmniArch (ICML 2025), which uses a Fourier encoder to move spatial field values to the frequency domain.
> 5. DPOT (ICML 2024), our main backbone, which is the only open‑source model in this family with more than 1B parameters and outperforms purely transformer‑based baselines such as MPP in their original paper.
>
> Secondly, although an FNO backbone provides direct access to frequency features, our main focus is on assigning each frequency band its own proper bottleneck dimension rather than strictly performing convolution in the frequency domain. This insight allows our F‑Adapter to extend naturally to non‑FFT architectures. On the pure transformer‑based **Poseidon** model, we estimate frequency energy for each Linear layer from adjacent‑token differences, and for each Conv2d layer we perform a local real 2‑D FFT on the convolution output to obtain an energy spectrum that guides the adapter’s weight generation. The adapter itself still operates in the native spatial domain. Capacities are allocated to bands according to their energy, following Eq(6) of our paper, which equips the model with frequency awareness. The accompanying table reports the resulting performance gains on 2D Shallow Water Equation Dataset.
> |Scheme on Poseidon|SWE-2D L2RE|%Param|
> |-|-|-|
> |Prompt Tuning|>1.0|0.07%|
> |LoRA|0.4010|2.07%|
> |RandLoRA|0.3134|2.07%|
> |Vanilla Adapter|0.6231|2.18%|
> |AdaLoRA|0.3756|2.32%|
> |HydraLoRA|0.3474|2.57%|
> |FiLM Adapter|0.4567|3.19%|
> |SVFT|0.6742|4.22%|
> |F-Adapter|0.4311|4.17%|
> |**F-LoRA**|**0.2746**|4.78%|
> |Full Fine-Tuning|0.1534|100%|
>
> We observe that adapters deliver strong results when the base model is an FNO, yet their effectiveness declines sharply on a transformer backbone. In contrast, LoRA and its variants demonstrate robust performance on transformer backbones, reflecting established best practices in fine-tuning LLMs. But our F‑Adapter still narrows this gap by significantly improving adapter performance on transformers. **Building on this insight, we introduce F‑LoRA: it preserves the frequency‑based capacity allocation of F‑Adapter while replacing the bottleneck MLP with LoRA‑style low‑rank linear updates.** F‑LoRA achieves SOTA performance across a broad suite of PEFT methods in this setting.
>
> **Reply to W2**
>
> You raise a valid point. In our paper, we define a Large Operator Model (LOM) as a pre‑trained foundational model that reaches the parameter scale commonly associated with “large models” and learns solution operators for PDEs. The ambiguity you noted comes from our wording: we meant only those LOMs built on the FNO backbone, not models using transformer‑based architectures. We will revise this sentence in the camera ready version to make it clear that “large‑scale operator models” in that sentence refers exclusively to FNO‑based LOMs.
>
> **Reply to Q1**
>
> While LoRA is not designed to improve performance, it serves as a viable option when memory and computational budgets are limited. Therefore, the goal of this work is to find a better parameter-efficient fine-tuning method that can achieve stronger performance under the same memory and computational constraints, rather than seeking improvements over the full-parameter fine-tuning method. **Please refer to Table 9 in the appendix of our paper.** When compared to LoRA, F‑Adapter achieves comparable FLOPs and per‑step inference time, while delivering superior performance.
>
> **Reply to Q2**
>
> SFT adapts a pre‑trained large model to a downstream task by updating all of its parameters so as to minimize the supervised loss on labeled data. PEFT pursues the same objective using labeled data but only introduces or updates a small set of extra parameters, while keeping the backbone weights fixed. In other words, SFT adjusts every model weight, whereas PEFT makes only minimal changes, making it a more resource friendly variant of supervised fine tuning. Both methods share the goal of maximizing task performance; their sole distinction lies in the fraction of parameters that are trainable. We acknowledge the relevance of Chen et al. (2024) [1], which explores data-efficient operator learning via unsupervised pretraining and in-context learning (ICL) in SciML. While their work focuses on reducing labeled data dependency through ICL, a non-fine-tuning approach, our study prioritizes PEFT for scenarios where task-specific labeled data exists but full parameter updates are computationally prohibitive. **As shown in Table 2 and Table 9, we have empirically compared the performance and computational cost of full fine tuning against several PEFT techniques.**

---

> ### Comment · Reviewer_dyWR · 2025-08-05
>
> I would like to thank the author for the detailed responses that addresses my concerns. I will maintain my score as 4.

---

> > ### Author Response · Authors · 2025-08-06
> >
> > We sincerely appreciate the reviewer’s time and positive evaluation. And we are pleased that our detailed responses have satisfactorily addressed your concerns.

---

### Official Review · Reviewer_G4t8 · 2025-07-03

**Clarity:** 3
**Significance:** 2
**Originality:** 3
**Rating:** 4
**Confidence:** 3

**Summary:**

This paper presents F-Adapter, a frequency-adaptive, parameter-efficient fine-tuning method specifically designed for pre-trained Large Operator Models (LOMs) in scientific machine learning. Unlike existing approaches such as LoRA, which are widely used in NLP and vision tasks, F-Adapter leverages the spectral structure of PDE solutions by allocating adapter capacity based on the distribution of energy across frequency bands. The authors provide both theoretical justifications 1) highlighting limitations of LoRA in Fourier-based architectures and 2) extensive empirical evaluations on 3D Navier–Stokes forecasting tasks. The proposed method consistently outperforms prior PEFT baselines while tuning fewer than 2% of the model parameters.

**Questions:**

1. Beyond FFT-Based models: Can F-Adapters be extended to non-Fourier architectures, such as graph-based operators or other models?

2. Sensitivity to adapter scheduling: How sensitive is performance to the bottleneck allocation schedule? Would a learned or dynamically adapted schedule perform better?

3. High-frequency regimes: While PDE solutions are often low-frequency dominated, some scenarios (e.g., turbulence, shocks) require high-frequency fidelity. How does F-Adapter perform in such regimes? Would be great to see comparison in those tasks.

**Ethical Concerns:**

["NO or VERY MINOR ethics concerns only"]

**Final Justification:**

The authors addressed my questions and concerns, and thus I am willing to raise my score.

**Limitations:**

yes

**Paper Formatting Concerns:**

No major formatting issues

**Quality:**

2

**Strengths And Weaknesses:**

Strengths:
1. Nice motivation. The paper offers a compelling analysis of why LoRA struggles in this domain, backed by provable lower bounds on approximation error and a universal approximation argument for adapters.
2. The frequency-adaptive allocation of adapter capacity is elegant and grounded in the spectral properties of PDEs, which are clearly laid out both empirically and theoretically.
3. Strong empirical results: across several challenging benchmarks, F-Adapter achieves state-of-the-art performance among PEFT methods, all with minimal parameter overhead.

Weakness:
1. Limited scope. The method and experiment mainly focus on a specific architecture (DPOT) and PDE family (Navier–Stokes). It would be helpful to understand how broadly the method applies to other operator learning settings, especially those not based on FFTs.
2. Hyperparamter and tuning cost. The dimension–frequency scheduling is manually set and fixed across tasks. It might be case by case and require manual tuning. An adaptive one will be better.

---

> ### Author Rebuttal · Authors · 2025-07-31
>
> Thanks for the thoughtful comments!
>
> **Reply to W1 & Q1**
>
> **Firstly, FFTs are still the cornerstone of most LOMs, much as Multi‑Head Attention is for Large Language Models.** Representative FFT‑based LOMs include
> 1. PreLowD (TMLR 2024), which relies on a factorized Fourier Neural Operator whose core loop is FFT, inverse FFT, and spectral convolution.
> 2. UPS (TMLR 2024), whose first stage applies FNO spectral blocks to map the physical field into a token sequence processed by a language‑model body.
> 3. CoDA‑NO (NeurIPS 2024), where on a uniform grid the key operators K, Q, V, M, and the integral operator I are all implemented with FNO components.
> 4. OmniArch (ICML 2025), which uses a Fourier encoder to move spatial field values to the frequency domain.
> 5. DPOT (ICML 2024), our main backbone, which is the only open‑source model in this family with more than 1B parameters and outperforms purely transformer‑based baselines such as MPP in their original paper.
>
> Secondly, although an FNO backbone provides direct access to frequency features, our main focus is on assigning each frequency band its own proper bottleneck dimension rather than strictly performing convolution in the frequency domain. This insight allows our F‑Adapter to extend naturally to non‑FFT architectures. On the pure transformer‑based **Poseidon** model, we estimate frequency energy for each Linear layer from adjacent‑token differences, and for each Conv2d layer we perform a local real 2‑D FFT on the convolution output to obtain an energy spectrum that guides the adapter’s weight generation. The PEFT adapter itself still operates in the native spatial domain. Capacities are allocated to bands according to their energy, following Eq(6) of our paper, which equips the model with frequency awareness. The accompanying table reports the resulting performance gains on 2D Shallow Water Equation Dataset.
> |Scheme on Poseidon|SWE-2D L2RE|%Param|
> |-|-|-|
> |Prompt Tuning|>1.0|0.07%|
> |LoRA|0.4010|2.07%|
> |RandLoRA|0.3134|2.07%|
> |Vanilla Adapter|0.6231|2.18%|
> |AdaLoRA|0.3756|2.32%|
> |HydraLoRA|0.3474|2.57%|
> |FiLM Adapter|0.4567|3.19%|
> |SVFT|0.6742|4.22%|
> |F-Adapter|0.4311|4.17%|
> |**F-LoRA**|**0.2746**|4.78%|
> |Full Fine-Tuning|0.1534|100%|
>
> We observe that adapters deliver strong results when the base model is an FNO, yet their effectiveness declines sharply on a transformer backbone. In contrast, LoRA and its variants demonstrate robust performance on transformer backbones, reflecting established best practices in fine-tuning LLMs. But our F‑Adapter still narrows this gap by significantly improving adapter performance on transformers. **Building on this insight, we introduce F‑LoRA: it preserves the frequency‑based capacity allocation of F‑Adapter while replacing the bottleneck MLP with LoRA‑style low‑rank linear updates.** F‑LoRA achieves SOTA performance across a broad suite of PEFT methods in this setting.
>
> **Reply to W2 & Q2**
>
> We would like to clarify that the hyperparameters that determine the model’s dimensionality may be difficult to make adaptive or learnable, since they fix the shape of weight tensors and cannot be updated during training. Nonetheless, we conducted extensive ablation studies on diverse hyperparameter settings using DPOT and F-Adapter on the 3D-Turbulance dataset.
>
> |p|r_min|r_max|B|L2RE|%Param|FLOPs(G)|B1 dim|B2 dim|B3 dim|B4 dim|B5 dim|B6 dim|
> |-|-|-|-|-|-|-|-|-|-|-|-|-|
> |2|4|16|4|0.4523|1.91%|548.5307|13|8|5|4|/|/|
> |2|4|16|6|0.4509|2.45%|548.7430|14|10|8|6|4|4|
> |2|8|32|4|0.4191|3.40%|555.3716|22|11|8|8|/|/|
> |1|16|32|4|0.4203|4.38%|556.8579|29|23|17|11|/|/|
> |4|16|64|4|0.3885|6.76%|569.9026|44|23|16|16|/|/|
> |1|16|64|4|0.4152|8.00%|572.8753|58|46|34|22|/|/|
>
> Results indicate that hyperparameters primarily influence performance by modulating adapter capacity allocation across frequency bands. This adjustment effectively governs the model's overall capacity. **Crucially, hyperparameters do not directly affect performance. Their impact is mediated through capacity allocation.** Consequently, selecting appropriate capacity based on computational resource constraints enables predictable performance outcomes. The magnitude of this impact remains relatively limited. This finding aligns with the insight presented in Table 1 of our original paper, which shows that adapters nearly obey the scaling law. Our Band-Specific Bottleneck Allocation framework (Eq 6) maintains robust generalization across diverse tasks, while hyperparameters retain flexibility to accommodate available computational resources.
>
> **Reply to Q3**
>
> Firstly, we have reported turbulence experiments in both Table 2 and Table 3. Now we supplement our study with a new set of comparative tests on the 2D shallow‑water equations (SWE-2D) and the 3D magnetohydrodynamic (MHD-3D) equations. In the SWE setting, phenomena such as hydraulic jumps and sharply non‑uniform interfaces generate shock‑like discontinuities and rich high‑wavenumber content, requiring high‑frequency fidelity for accurate capture. The MHD equations represent a class of PDEs that are characterized by broadband spectral behavior. Moreover, we only extracted 24 trajectories for training, aiming to evaluate the performance under extreme data scarcity conditions.
>
> |Scheme on DPOT|SWE-2D L2RE|SWE-2D %Param|MHD-3D L2RE|MHD-3D %Param|
> |-|-|-|-|-|
> |AdaLoRA|0.1061|0.70%|1.0022|0.69%|
> |HydraLoRA|0.0956|0.88%|0.9440|0.85%|
> |Prompt Tuning|0.1050|0.11%|0.9950|1.03%|
> |Vanilla Adapter|0.0902|0.48%|0.7226|1.16%|
> |FiLM Adapter|0.0162|0.57%|0.7593|1.30%|
> |RandLoRA|0.1568|1.05%|0.9800|1.36%|
> |LoRA|0.1081|1.40%|0.9845|1.37%|
> |F-Adapter(Ours)|**0.0116**|1.24%|**0.6341**|1.91%|
> |SVFT|0.0975|0.84%|1.0004|2.31%|
> |Full Fine-Tuning|0.0023|100%|0.4190|100%|
>
> Our results demonstrate that F‑Adapter still achieves SOTA. Intuitively, this stems from our frequency‑adaptive allocation of model capacity. **Rather than neglecting high‑frequency features, we partition the spectrum so that all frequencies are processed more efficiently and effectively.** F‑Adapter combines three complementary mechanisms: wide channels for low‑frequency components, lightweight residual connections for high‑frequency components, and a nonlinear universal approximator to concentrate limited parameters on the most critical features. High‑frequency information, although important, is energetically sparse and can be captured with small bottleneck modules. These modules both preserve essential details and impose an implicit regularization. Consequently, F‑Adapter excels in turbulent flows and other scenarios dominated by high‑frequency dynamics.

---

### Note · Authors · 2025-08-12

We thank all reviewers for their thoughtful comments and efforts. We have responded to each reviewer individually and addressed all concerns point by point. Below is a concise summary of what we mainly have solved during rebuttal period.

* **Transfer to other PDE settings.** We added experiments on SWE-2D and on MHD-3D under data-scarce conditions. F-Adapter attains SOTA performance in these settings.

* **Transfer to other LOM architectures.** On Poseidon with SWE-2D, across a broad set of PEFT baselines, F-Adapter substantially improves the adapter family. We migrate the frequency based capacity allocation principle to a transformer friendly variant, F-LoRA, which achieves SOTA results.

* **Hyperparameter sensitivity.** We added experiments with diverse hyperparameter settings. The results indicate that hyperparameters do not directly affect performance; their effect is mediated through how capacity is allocated across frequency bands.

* **Velocity spectrum analysis.** We compared spectral metrics between F-Adapter and multiple PEFT baselines. F-Adapter accurately captures the energy spectrum.

* **Visual comparisons.** We carried out visual comparisons of predicted and ground truth velocity fields using slice plots and isosurfaces. F-Adapter preserves filament like vortical structures while maintaining overall flow fidelity. The results will be well presented at CRV.

Finally, we would like to emphasize that **this work is the first to explore PEFT for scientific machine learning** and establishes frequency-based capacity allocation as an effective paradigm for this domain. **If accepted, this work could provide valuable insights to the community.**

---

### Decision · Program_Chairs · 2025-09-17

**Decision:**

Accept (poster)

**Comment:**

This paper introduces F-Adapter, a frequency-adaptive PEFT method for large Fourier Neural Operator models. By allocating adapter capacity according to the spectral energy of PDE solutions, it prioritizes low-frequency modes while limiting high-frequency capacity.
After reviewing the rebuttal, the reviewers reached a consensus that the paper demonstrates a principled design, rigorous theoretical grounding, and effective empirical validation with extensive ablations. Most experimental concerns raised earlier were addressed. The authors are encouraged to incorporate the references suggested by the reviewers and discuss them in the related works.
Overall, the AC recommends acceptance.